# Optimistic Online Learning in Symmetric Cone Games

**Anas Barakat**  *anas_barakat@sutd.edu.sg*
*Singapore University of Technology and Design*

**Wayne Lin**  *wayne_lin@sutd.edu.sg*
*Singapore University of Technology and Design*

**John Lazarsfeld**  *john_lazarsfeld@sutd.edu.sg*
*Singapore University of Technology and Design*

**Antonios Varvitsiotis**  *antonios@sutd.edu.sg*
*Singapore University of Technology and Design*
*Centre for Quantum Technologies, National University of Singapore*
*Archimedes/Athena Research Center*

**Reviewed on OpenReview:** *https://openreview.net/forum?id=RQjvE2o3o6*

## Abstract

We introduce symmetric cone games (SCGs), a broad class of multi-player games where each player's strategy lies in a generalized simplex (the trace-one slice of a symmetric cone). This framework unifies a wide spectrum of settings, including normal-form games (simplex strategies), quantum games (density matrices), and continuous games with ball-constrained strategies. It also captures several structured machine learning and optimization problems, such as distance metric learning and Fermat–Weber facility location, as two-player zero-sum SCGs. To compute approximate Nash equilibria in two-player zero-sum SCGs, we propose a single online learning algorithm: Optimistic Symmetric Cone Multiplicative Weights Updates (OSCMWU). Unlike prior methods tailored to specific geometries, OSCMWU provides closed-form updates over any symmetric cone and achieves a $\tilde{\mathcal{O}}(1/\epsilon)$ iteration complexity for computing $\epsilon$-saddle points. Our analysis builds on the Optimistic Follow-the-Regularized-Leader framework and hinges on a key technical contribution: We prove that the symmetric cone negative entropy is strongly convex with respect to the trace-one norm. This result extends known results for the simplex and spectraplex to all symmetric cones, and may be of independent interest.

## 1 Introduction

Problems such as distance metric learning (e.g. Weinberger & Saul (2009)), adversarial training of quantum generative models (Dallaire-Demers & Killoran, 2018; Chakrabarti et al., 2019), and facility location optimization (Brimberg, 1995; Xue & Ye, 1997) may seem unrelated at first glance. Yet, all of them can be formulated as two-player zero-sum games where each player optimizes over a structured, convex strategy space. These strategy spaces take a diversity of forms (probability simplices, trace-one positive semidefinite (PSD) matrices, and Euclidean balls) reflecting different algebraic or geometric constraints.

While this shared structure suggests the potential for unified solution methods, existing algorithms remain highly fragmented, often tailored to specific geometries in special structured problems. For instance, distance metric learning can be solved using the Frank-Wolfe algorithm (Ying & Li, 2012) or Nesterov's smoothing algorithm (Nesterov, 2007); quantum zero-sum games can be tackled using the Matrix Multiplicative Weights Update algorithm (Jain & Watrous, 2009; Jain et al., 2022); the celebrated Fermat-Weber facility location problem can be solved using interior point methods (Xue & Ye, 1997). This fragmented landscape of

algorithms and analyses calls for the design of broadly applicable algorithms for equilibrium learning in structured games.

In this work, we present a unified algorithmic framework for equilibrium learning in such problems by introducing a rich and expressive class of games we call Symmetric Cone Games (SCGs). In SCGs, each player's strategy set is a generalized simplex—the set of trace-one elements in a symmetric cone, i.e., the cone of squares of a Euclidean Jordan Algebra (EJA). This abstraction includes many familiar structures including the simplex (trace-one slice of the nonnegative orthant), the spectraplex (trace-one slice of the PSD cone), the second-order cone (trace-one $\ell^2$ ball), and products of these cones. As a consequence, Symmetric Cone Games notably generalize several well-known classes: Normal-form games, where strategies lie in simplices; Quantum games, where strategies are density matrices (PSD, trace-one); continuous games with Euclidean geometry structure where strategies live in $\ell^2$ balls or second-order cones.

While we define SCGs in the general multi-player setting, we focus on the foundational case of two-player zero-sum SCGs, which unify and extend a variety of structured optimization problems in machine learning and game theory. Concretely, we study the following min-max problem:

$$\min_{x \in \Delta_{\mathcal{K}_1}} \max_{y \in \Delta_{\mathcal{K}_2}} f(x, y), \tag{1}$$

where $\Delta_{\mathcal{K}1}, \Delta_{\mathcal{K}_2}$ are generalized simplices over symmetric cones $\mathcal{K}_1, \mathcal{K}_2$ respectively, and $f$ is convex-concave. This general formulation accommodates rich structured domains such as PSD cones (spectral information), second-order cones (geometric norms), and their products. We illustrate this generality through concrete applications including distance metric learning (as a simplex–spectraplex game), facility location (as a second-order cone game).

To solve such problems, we introduce a new online first-order learning algorithm—Optimistic Symmetric Cone Multiplicative Weights Update (OSCMWU)—for computing approximate saddle points in SCGs. This closed-form update method operates over any symmetric cone, does not require a Euclidean projection onto the symmetric cone and can be run online and independently by each player. OSCMWU recovers the Optimistic Multiplicative Weights Update (OMWU) algorithm for simplex games (Syrgkanis et al., 2015) and its matrix variants for spectraplex domains (Vasconcelos et al., 2023) as particular cases. In two-player SCGs, the algorithm achieves $\mathcal{O}(1/\varepsilon)$ iteration complexity for reaching an $\varepsilon$-saddle point, improving over the prior $\mathcal{O}(1/\varepsilon^2)$ bound for vanilla SCMWU (Canyakmaz et al., 2023). Conceptually, OSCMWU is an instance of Optimistic Follow-the-Regularized-Leader (OFTRL) with a cone-specific entropy regularizer. A key technical ingredient in our analysis is the following strong convexity result: We prove that the symmetric cone negative entropy is strongly convex with respect to the trace-one norm. This generalizes known results for the simplex and spectraplex to arbitrary symmetric cones, leveraging the algebraic structure of Euclidean Jordan algebras (Faraut & Korányi, 1994). We believe this result and its proof, based on a novel data processing inequality, may be of independent interest to the learning and optimization communities.

We illustrate the versatility of our framework by applying OSCMWU to two representative SCGs, showcasing its generality, simplicity, and robust convergence across different cone geometries.

In summary, our main contributions are as follows:

- **A unifying class of structured convex games.** We introduce *Symmetric Cone Games* (SCGs), a general class that subsumes numerous existing settings under a single abstraction, including normal-form, quantum, and continuous games with Euclidean ball strategy spaces.

- **An online learning algorithm.** We propose *OSCMWU*, an optimistic algorithm that operates over *any* symmetric cone and can be run independently by each player in an SCG. The method yields closed-form updates via the exponential map and is well-suited for streaming, adversarial, or interactive learning scenarios in modern machine learning. In two-player SCGs, OSCMWU guarantees a $\mathcal{O}(1/\varepsilon)$ iteration complexity to reach an $\varepsilon$-saddle point. This result improves over the $\mathcal{O}(1/\varepsilon^2)$ result established in Canyakmaz et al. (2023).

- **A strong convexity result.** We establish that the symmetric cone negative entropy is strongly convex with respect to the trace-one norm. This generalizes classical results for the simplex and spectraplex to all symmetric cones and plays a key role in the regret analysis of our method.

- **Applications.** We apply OSCMWU to solve SCGs arising in structured prediction (distance metric learning), spatial optimization problems (facility location), and hybrid cone optimization, demonstrating the versatility of our unified framework beyond classical game settings.

Our theoretical results point toward a broader unification of structured online game-theoretic learning methods and their application to diverse domains. Proofs and extended related work are deferred to the appendix.

## 2 Preliminaries: Euclidean Jordan Algebras, Symmetric Cones, and Entropy

In this section, we introduce our notations and define symmetric cones. We rely on the formalism of Euclidean Jordan algebras that we briefly review. We refer the reader to Appendix B for additional background and examples, and Faraut & Korányi (1994) for a more detailed exposition.

**(Euclidean) Jordan algebras.** Let $\mathcal{J}$ be a finite dimensional vector space equipped with a bilinear product $\circ : \mathcal{J} \times \mathcal{J} \to \mathcal{J}$. The pair $(\mathcal{J}, \circ)$ forms a Jordan algebra if for all $x, y \in \mathcal{J}, x \circ y = y \circ x$ and $x^2 \circ (x \circ y) = x \circ (x^2 \circ y)$ where $x^2 = x \circ x$. We assume throughout this paper that the Jordan algebra $(\mathcal{J}, \circ)$ (simply denoted by $\mathcal{J}$) has an identity element, i.e. there exists $e \in \mathcal{J}$ s.t. $e \circ x = x \circ e = x$ for every $x \in \mathcal{J}$. A Jordan algebra over $\mathbb{R}$ is said to be *Euclidean* if it is equipped with an associative inner product $(\cdot, \cdot)$, i.e. for all $x, y, z \in \mathcal{J}, (x \circ y, z) = (y, x \circ z)$. Throughout this work, we will simply denote a Euclidean Jordan algebra (EJA in short) $(\mathcal{J}, \circ, (\cdot, \cdot))$ by $\mathcal{J}$.

**Symmetric cones.** A cone $\mathcal{K}$ in an inner product space is said to be *symmetric* if it is self-dual and homogeneous. Notably, *any* symmetric cone is the cone of squares $\{x \circ x : x \in \mathcal{J}\}$ of some EJA $\mathcal{J}$ (see Faraut & Korányi (1994) Theorem III.3.1). Table 2 in Appendix B.3 summarizes some relevant examples.

**Jordan frames and spectral decomposition.** An element $p \in \mathcal{J}$ s.t. $p \circ p = p$ is called an idempotent in $\mathcal{J}$. It is a *primitive* idempotent if it is nonzero and cannot be written as a sum of two nonzero idempotents. A Jordan frame is a finite set $\{q_1, \ldots, q_r\}$ of primitive idempotents in $\mathcal{J}$ s.t. $q_i \circ q_j = 0$ if $i \neq j$ and $\sum_{i=1}^{r} q_i = e$ where $e$ is the unit element of $\mathcal{J}$ and $r$ is the rank of the EJA, see Appendix B.2. If $\mathcal{J}$ is an EJA with rank $r$, then for every $x \in \mathcal{J}$, there exists a Jordan frame $\{q_1, \cdots, q_r\}$ and $\lambda_1, \cdots, \lambda_r \in \mathbb{R}$ s.t. $x = \sum_{i=1}^{r} \lambda_i q_i$. The scalars $(\lambda_i)_{1 \leq i \leq r}$ are called the eigenvalues of $x$, and are uniquely determined up to multiplicities. (See e.g. Faraut & Korányi (1994) Theorem III.1.2 for this so-called type-II spectral decomposition.) This decomposition generalizes the usual spectral decomposition of a Hermitian matrix into rank-one projectors.

**Trace, inner product, dual norms, trace $p$-norms.** The trace map $\text{tr} : \mathcal{J} \to \mathbb{R}$ is equal to the sum of eigenvalues, i.e., $\text{tr}(x) = \sum_{i=1}^{r} \lambda_i$, for every $x \in \mathcal{J}$. The bilinear mapping $(x, y) \mapsto \text{tr}(x \circ y)$ is positive definite, symmetric and associative (see e.g. Faraut & Korányi (1994), Prop. III.4.1). We use the notation $\langle x, y \rangle = \text{tr}(x \circ y)$ for this canonical inner product. If $E$ is a finite-dimensional real space, we denote its dual by $E^*$ and we will write $\langle y, x \rangle$ for the pairing between $x \in E$ and $y \in E^*$. For any given norm $\| \cdot \|$ defined on the space $E$, we use the standard definition of the dual norm $\| \cdot \|_*$ for every $v \in E^*$ by $\|v\|_* = \sup_{\|u\| \leq 1} \langle u, v \rangle$ where $\langle \cdot, \cdot \rangle$ is the previously defined pairing. Finally, for any $x \in \mathcal{J}$, its trace $p$-norm is defined by $\|x\|_{\text{tr},p} := \sum_{i=1}^{r} |\lambda_i(x)|^p$ when $0 < p < \infty$ and $\|x\|_{\text{tr},\infty} := \max_{1 \leq i \leq r} |\lambda_i(x)|$ when $p = \infty$, where $\{\lambda_i(x)\}_{1 \leq i \leq r}$ are the eigenvalues of $x$. The trace-$p$ norms were first introduced and proven to be norms for all $p \in [1, \infty]$ by Tao et al. (2014), and Gowda (2019) proved that the trace-$p$ and trace-$q$ norms are dual to each other for all $p, q \in [1, \infty]$ satisfying $\frac{1}{p} + \frac{1}{q} = 1$.

**Symmetric cone negative entropy.** Given the EJA $\mathcal{J}$ and its cone of squares $\mathcal{K}$, the *negative entropy* mapping $\Phi_{\text{ent}} : \text{int}(\mathcal{K}) \to \mathbb{R}$ is defined for every $x \in \text{int}(\mathcal{K})$ by:

$$\Phi_{\text{ent}}(x) = \text{tr}(x \circ \ln x) = \sum_{i=1}^{r} \lambda_i \ln \lambda_i, \tag{SCNE}$$

where $x = \sum_{i=1}^r \lambda_i q_i \in \text{int}(\mathcal{K})$ is the spectral decomposition of $x$ and $\ln : \text{int}(\mathcal{K}) \to \mathcal{J}$ is the Löwner extension of the scalar logarithm function defined by $\ln x = \sum_{i=1}^r \ln(\lambda_i) q_i$ for any $x \in \text{int}(\mathcal{K})$. Note that the last expression is well defined since $\lambda_i > 0$ for every $1 \le i \le r$ as $x \in \text{int}(\mathcal{K})$. Similarly, the exponential mapping $\exp : \mathcal{J} \to \text{int}(\mathcal{K})$ is defined by $\exp(x) = \sum_{i=1}^r \exp(\lambda_i) q_i$.

## 3 Symmetric Cone Games

Symmetric cone games (SCGs) are multi-player games in which each player selects strategies from a generalized simplex, defined as the trace-one slice of a symmetric cone. This structure captures a broad class of structured learning and optimization problems—including classical and quantum games, metric learning, and convex programs over cones—under a unified theoretical framework. Formally, if $(\mathcal{J}, \circ)$ is an EJA of rank $r$ and $\mathcal{K}$ its cone of squares, we define the generalized simplex:

$$\Delta_{\mathcal{K}} := \{x \in \mathcal{K} : \text{tr}(x) = 1\}, \tag{2}$$

i.e. the trace-one slice of $\mathcal{K}$. Note that $\Delta_{\mathcal{K}}$ is a convex compact set. Table 1 illustrates how SCGs subsume well-known domains including normal-form games, quantum games, and games with Euclidean geometry.

Table 1: Examples of Generalized simplices

| Symmetric Cone $\mathcal{K}$ | Generalized Simplex $\Delta_{\mathcal{K}}$ | |
|---|---|---|
| Nonnegative orthant $\mathbb{R}^n_+$ | Simplex | $\Delta^{n-1} = \{x \in \mathbb{R}^n_+ : \sum_{i=1}^n x_i = 1\}$ |
| Real PSD symmetric matrices $\mathbb{S}^n_+$ | Spectraplex | $\Delta_{\mathbb{S}^n_+} = \{X \in \mathbb{S}^n_+ : \text{Tr}(X) = 1\}$ |
| PSD Hermitian matrices $\mathbb{H}^n_+$ | Spectraplex[1] | $\Delta_{\mathbb{H}^n_+} = \{X \in \mathbb{H}^n_+ : \text{Tr}(X) = 1\}$ |
| Second-order cone $\mathbb{L}^n_+$ | Ball[2] | $\Delta_{\mathbb{L}^n_+} = \{(\frac{1}{2}, x) \in \mathbb{R}^n : \|x\|_2 \le \frac{1}{2}\}^*$ |

[1] This spectraplex is the set of density matrices and is commonly used in quantum computation and in quantum games, see e.g. Gutoski & Watrous (2007); Jain & Watrous (2009). [2] We call it a ball here with a slight abuse since it is the set of $(\frac{1}{2}, x)$ and not only $x$.

We consider a game setting with a finite number of players $\mathcal{N} := \{1, \cdots, N\}$. Each player $i \in \mathcal{N}$ selects a strategy $x_i$ from a generalized simplex $\Delta_{\mathcal{K}_i}$ associated to a symmetric cone $\mathcal{K}_i$ (which is a closed convex subset of a finite-dimensional EJA $\mathcal{J}_i$). We denote the space of all joint strategies by $\mathcal{X} := \prod_{i \in \mathcal{N}} \Delta_{\mathcal{K}_i} \subseteq \mathcal{J} := \prod_{i \in \mathcal{N}} \mathcal{J}_i$. We recall the standard notation $x = (x_i, x_{-i})$ for any joint strategy $x \in \mathcal{X}$. Each player $i \in \mathcal{N}$ has a payoff (or utility, reward) function $u_i : \mathcal{X} \to \mathbb{R}$ which will be assumed to be concave and differentiable[1] w.r.t. its $i$th variable. Under this assumption, we introduce the payoff vector as follows: $\forall x \in \mathcal{X}, m(x) = (m_i(x))_{i \in \mathcal{N}}, m_i(x) = \nabla_{x_i} u_i(x_i, x_{-i}), \forall i \in \mathcal{N}$.

At each time step $t$, each player $i \in \mathcal{N}$ observes their payoff vector $m_i^t = m_i(x^t)$ where $x^t \in \mathcal{X}$ is the joint strategy of the players. Then player $i$ receives the payoff $u_i(x^t)$ and computes their next strategy $x_i^{t+1} \in \Delta_{\mathcal{K}_i}$.

The playerwise regret associated to a sequence $(x_i^t)$ of player $i$'s strategies after $T$ steps is given by:

$$r_i(T) := \sup_{x \in \Delta_{\mathcal{K}_i}} \sum_{t=1}^T u_i(x, x_{-i}^t) - u_i(x_i^t, x_{-i}^t), \quad \forall i \in \mathcal{N}. \tag{3}$$

We assume that each player's payoff vector is Lipschitz-continuous under the trace norm.

**Assumption 1.** $\forall x, y \in \mathcal{X}, \forall i \in \mathcal{N}, \exists L_i \ge 0, \|m_i(x) - m_i(y)\|_{\text{tr},\infty} \le L_i \|x - y\|_{\text{tr},1}$.

This smoothness assumption on the payoff vectors is common in the literature, see e.g. Farina et al. (2022a); Mertikopoulos et al. (2024) (Assumption 1). Notably, we observe that if the game is multilinear and the absolute value of the utility $u_i(x)$ is bounded by $L_i$, then Assumption 1 holds (see Proposition 23 in Appendix F). In the next section, we discuss further examples of SCGs where Assumption 1 is satisfied.

---

[1]For differentiability, the utility functions can be seen to be defined over the EJA $\mathcal{J}$ which is an open set.

### 3.1 Examples of SCGs

We now discuss examples that illustrate how SCGs instantiate key classes of games.

**Example 1** (Finite normal-form games). Each player $i \in \mathcal{N}$ selects an action from a finite set $\mathcal{A}_i$ and their payoff function $u_i : \mathcal{A} \to \mathbb{R}$ is defined over the product space $\mathcal{A} := \prod_{i \in \mathcal{N}} \mathcal{A}_i$.[2] Each player may choose actions randomly according to a probability distribution $x_i = (x_{i,a_i})_{a_i \in \mathcal{A}_i} \in \Delta(\mathcal{A}_i)$ where $\Delta(\mathcal{A}_i)$ is the simplex. The payoff functions are extended as multilinear functions $u_i : \prod_{i \in \mathcal{N}} \Delta(\mathcal{A}_i) \to \mathbb{R}$ defined by $u_i(x) = \sum_{a \in \mathcal{A}} x_a u_i(a)$ where $x_a = \prod_{i \in \mathcal{N}} x_{i,a_i}$ is the probability of the joint action $a = (a_1, \cdots, a_n)$. Note that in this setting $m_i(x) = \nabla_{x_i} u_i(x) = (u_i(a_i, x_{-i}))_{a_i \in \mathcal{A}_i}$ where $u_i(a_i, x_{-i}) := \sum_{a_{-i} \in \mathcal{A}_{-i}} x_{a_{-i}} u_i(a_i, a_{-i})$ and $\mathcal{A}_{-i} := \prod_{j \in \mathcal{N}, j \neq i} \mathcal{A}_j$. Assumption 1 is satisfied for bounded payoff functions (see e.g. the proof of Theorem 4 in Syrgkanis et al. (2015)).

**Example 2** (Convex games with ball strategy sets). Consider the second-order (Lorentz) cone $\mathbb{L}_+^n = \{x = (x_1, \bar{x}) \in \mathbb{R} \times \mathbb{R}^{n-1} : \|\bar{x}\|_2 \leq x_1\}$. The associated generalized simplex (see Table 1) is the affine slice $\Delta_{\mathbb{L}_+^n} = \{(\frac{1}{2}, \bar{x}) \in \mathbb{R}^n : \|\bar{x}\|_2 \leq \frac{1}{2}\}$ which is (via projection onto $\bar{x}$) affinely isomorphic to the Euclidean $\ell_2$ ball $\{\bar{x} \in \mathbb{R}^{n-1} : \|\bar{x}\|_2 \leq \frac{1}{2}\}$. A convex game with $\ell_2$ ball constraints is then specified by strategy sets $\mathcal{X}_i = \Delta_{\mathbb{L}_+^n}$ and payoff functions $u_i : \mathcal{X} := \prod_{j=1}^N \mathcal{X}_j \to \mathbb{R}$ that are concave in $x_i$ for each fixed $x_{-i}$ and continuously differentiable on $\mathcal{X}$ (see, e.g., Rosen (1965) for concave games). We provide two concrete examples of games with Euclidean $\ell_2$ ball strategy sets: (a) Fixed-budget vote allocation in social choice, where an agent allocates a vote vector $x_i \in \mathbb{R}^{d_i}$ subject to a quadratic budget constraint $\|x_i\|_2^2 \leq B_i$ (see, e.g., Georgescu et al. (2024)); (b) Beamforming in wireless communications, where each transmitter chooses a beamforming vector $x_i$ under a power constraint $\|x_i\|_2^2 \leq P_i$ (see, e.g., Larsson & Jorswieck (2008, section III)). More generally, Euclidean balls are canonical compact convex strategy sets, a standard assumption in convex game theory; see, e.g., Facchinei & Kanzow (2010).

**Example 3** (PSD matrix games). In this case, each player controls a PSD matrix variable (e.g. a signal covariance matrix) and each strategy space is the set of trace-one PSD matrices ($\Delta_{\mathbb{S}_+^n}$). These matrix games find applications in wireless communication networks for the competitive maximization of mutual information in interfering networks (Arslan et al., 2007; Scutari et al., 2008; Mertikopoulos & Moustakas, 2015; Majlesinasab et al., 2019).

**Example 4** (Quantum games). The strategies of the players are quantum states represented by density matrices, i.e. elements of the spectraplex $\Delta_{\mathbb{H}_+^n}$, and the utility is given by the expected value of a measurement on the joint state, i.e., the inner product of a Hermitian observable with an element in the tensor product space. If the players are playing separately, their joint state is a product state, and the utility function is multilinear in each player's individual strategy space. We refer the reader to Gutoski & Watrous (2007); Jain & Watrous (2009); Vasconcelos et al. (2023); Lin et al. (2024).

### 3.2 Representative Applications of SCGs

We now illustrate how diverse problems across machine learning and optimization fit naturally within our framework of SCGs. Each of the following examples is a convex-concave min-max game over generalized simplices derived from symmetric cones.

**Application 1: Distance Metric Learning via Spectraplex Games.** Distance metric learning aims to learn a Mahalanobis distance that pulls similar examples closer while pushing dissimilar ones apart. Consider a dataset $\{x_i\}_{1 \leq i \leq N}$ where $x_i \in \mathbb{R}^d$. For any pair $(i, j)$, we define the matrix $X_{i,j} := (x_i - x_j)(x_i - x_j)^T \in \mathbb{R}^{d \times d}$. The Mahalanobis distance induced by any PSD matrix $M \in \mathbb{S}_+^d$ is defined by $d_M^2(x_i, x_j) := (x_i - x_j)^T M (x_i - x_j) = \langle X_{i,j}, M \rangle$ for any pair $(x_i, x_j)$. Denote by $\mathcal{S}$ and $\mathcal{D}$ the index sets of similar and dissimilar pairs respectively, i.e. $(i, j) \in \mathcal{S}$ if $(x_i, x_j)$ is a pair of similar data points. Ying & Li (2012) propose to maximize the minimal squared distances between dissimilar pairs while controlling the

---

[2]We use the same notation $u_i$ as in the previous section with a slight abuse of notation, to be consistent with the standard payoff notation for normal-form games in game theory.

sum of squared distances between similar pairs:

$$\max_{M \in \mathbb{S}_+^d} \quad \min_{\tau \in \mathcal{D}} \langle X_\tau, M \rangle$$
$$\text{s.t.} \quad \langle X_{\mathcal{S}}, M \rangle \leq 1, \tag{4}$$

where $X_{\mathcal{S}} := \sum_{(i,j) \in \mathcal{S}} X_{i,j} \in \mathbb{S}_+^d$. Using duality, this problem is shown in Ying & Li (2012) (Theorem 1) to be equivalent to the convex–concave saddle-point problem:

$$\min_{x \in \Delta^{D-1}} \max_{Y \in \Delta_{\mathbb{S}_+^d}} \left\langle \sum_{\tau \in \mathcal{D}} x_\tau \tilde{X}_\tau, Y \right\rangle, \tilde{X}_\tau := X_S^{-1/2} X_\tau X_S^{-1/2}, \tag{5}$$

the matrix $X_{\mathcal{S}}$ is supposed to be invertible, $\Delta^{D-1} = \left\{ u \in \mathbb{R}^D : u_\tau \geq 0, \quad \sum_{\tau \in \mathcal{D}} u_\tau = 1 \right\}, D := |\mathcal{D}|$ is the number of pairs. This is a SCG over a simplex and a spectraplex. More broadly, in simplex-spectraplex games player 1 plays a probability simplex vector $x$ and player 2 plays a real symmetric, trace-1, PSD matrix $Y$, giving rise to the (bi-affine) min-max game:

$$\min_{x \in \Delta^{m-1}} \max_{Y \in \Delta_{\mathbb{S}_+^d}} f(x, Y) := \langle Y, \mathcal{A}(x) \rangle + \langle b, x \rangle + \langle C, Y \rangle, \tag{6}$$

where $\mathcal{A} : \mathbb{R}^m \to \mathbb{S}^d$ is the linear map given by $\mathcal{A}(x) = \sum_{i=1}^m x_i A_i$ for some $A_i \in \mathbb{S}^d$. Problem (6) captures a wide range of problems including some constant trace semi-definite programming problems and maximum eigenvalue minimization problems.

**Application 2: Facility Location Optimization via Second-Order Cone Games.** The sum-of-norms problem, also known as the Fermat–Weber location problem, arises in transportation, logistics, and communications (see e.g. Brimberg (1995); Xue & Ye (1997) and the references therein). It seeks to minimize the sum of Euclidean distances (or residual norms) to given targets. Its general form is:

$$\min_{x \in C} \left\{ g(x) := \sum_{i=1}^p \|A_i x - b_i\|_2 \right\}, \tag{7}$$

where $C := \{x \in \mathbb{R}^d : \|x\|_2 \leq R\}, \| \cdot \|_2$ is the standard Euclidean norm of $\mathbb{R}^d$ and for all $i \in \{1, \cdots, p\}, A_i \in \mathbb{R}^{m \times d}, b_i \in \mathbb{R}^m$.

We now reformulate problem (7) as a min-max problem involving second-order cone decision variables. We introduce a few useful notations. Define $\bar{A}_i := \left(0, A_i^T\right)^T \in \mathbb{R}^{(m+1) \times d}$ and $\bar{b}_i := \left(0, b_i^T\right)^T \in \mathbb{R}^{m+1}$. Now, recall that for any $(s, x) \in \mathbb{L}^{d+1}$, the two eigenvalues of $x$ are $s \pm \|x\|_2$ and the largest eigenvalue is therefore $s + \|x\|_2$. Now, using this fact together with the variational characterization of the maximal eigenvalue[3], we can write, by further introducing $\bar{A} := \left(\bar{A}_1^T, \cdots, \bar{A}_p^T\right)^T$ and $\bar{b} := \left(\bar{b}_1^T, \cdots, \bar{b}_p^T\right)^T \in \mathcal{J}^p$, the objective function as

$$g(x) = \sum_{i=1}^p \lambda_{\max}(\bar{A}_i x - \bar{b}_i) = \max_{y \in \Delta_{\mathcal{K}}^p} \langle y, \bar{A}x - \bar{b} \rangle_{\mathcal{J}^p}, \tag{8}$$

where $\mathcal{J} = \mathbb{L}^{d+1}$ and $\mathcal{J}^p$ is the product EJA. We now observe that $x \in C$ is equivalent to $(1/2, x/(2R)) \in \Delta_{\mathbb{L}_+^d} = \{(1/2, x) \in \mathbb{R}^{d+1} : \|x\| \leq 1/2\}$.[4] Combining this with (8) and using the change of variable $\tilde{x} = x/(2R)$, we obtain that problem (7) is equivalent to:

$$\min_{\bar{x} = (1/2, \tilde{x}) \in \Delta_{\mathbb{L}_+^{d+1}}} \max_{y \in \prod_{i=1}^p \Delta_{\mathbb{L}_+^{d+1}}} f(\bar{x}, y), \tag{9}$$

where $f(\bar{x}, y) := \langle \tilde{A} \bar{x}, y \rangle - \langle \bar{b}, y \rangle, \tilde{A} := 2R(0 \quad \bar{A})$. This min-max formulation fits directly within the SCG framework, where both player strategy sets are generalized simplices derived from second-order cones.

---

[3]In other words, $\lambda_{\max}$ is the support function of the generalized simplex, here: $\lambda_{\max}(x) = \max_{y \in \Delta_{\mathbb{L}_+^d}} \langle x, y \rangle$.

[4]Recall here that $\Delta_{\mathbb{L}_+^d}$ is the trace-one slice of the second-order cone $\mathbb{L}_+^d$ as $\text{tr}((t, x)) = 2t$ for any $(t, x) \in \mathbb{L}_+^d$.

# 4   Optimistic Symmetric Cone Multiplicative Weights Updates

In this section, we present our Optimistic Symmetric Cone Multiplicative Weights Update (OSCMWU) algorithm. The iterates of this algorithm evolve in a generalized simplex. We aim for a single algorithm that applies uniformly across all symmetric cone domains (e.g., simplices, spectraplexes, $l^2$-balls, etc.), avoiding projections and offering closed-form updates via the exponential map. To our knowledge, no such general-purpose algorithm has been proposed before. First we describe the online convex optimization (OCO) setting to set the stage (see e.g. Shalev-Shwartz (2012)).

**Online symmetric cone optimization.** At each time step $t$, a decision maker chooses an element of the generalized simplex $\Delta_\mathcal{K}$, i.e. a trace-one element $x^t$ of a symmetric cone $\mathcal{K}$ and receives a payoff $f^t(x^t)$ where $f^t : \Delta_\mathcal{K} \to \mathbb{R}$ is a concave differentiable gain function.[5] The decision maker then receives as feedback the vector $m^t := \nabla f^t(x^t)$. Note that $m^t \in \mathcal{J}^* = \mathcal{J}$, where $\mathcal{J}$ is $\mathcal{K}$'s associated EJA. This setting has been recently considered in Canyakmaz et al. (2023) in the particular case of linear payoffs, i.e where the payoff function $f^t$ is linear: for all $t \geq 0, f^t(x^t) = \langle m^t, x^t \rangle$ where $m^t \in \mathcal{J}$ is the observed payoff vector.

**Optimistic Follow The Regularized Leader.** In view of our analysis, we will present our optimistic multiplicative update algorithm as an instance of the general framework of Optimistic Follow-The-Regularized Leader (OFTRL) in online learning. Consider any strongly convex regularizer $\Phi : \text{int}(\mathcal{K}) \to \mathbb{R}$. As introduced and studied in e.g. Rakhlin & Sridharan (2013a;b); Syrgkanis et al. (2015), (OFTRL) with step size $\eta > 0$ can be written as follows over the generalized simplex $\Delta_\mathcal{K}$ with $x^0 = \underset{x \in \Delta_\mathcal{K}}{\text{argmin}}\, \Phi(x)$ and for all $t \geq 0$,

$$x^{t+1} = \underset{x \in \Delta_\mathcal{K}}{\text{argmax}} \left\{ \eta \left\langle \sum_{k=1}^{t} m^k + \tilde{m}^{t+1}, x \right\rangle - \Phi(x) \right\} \qquad \text{(OFTRL)}$$

where $(\tilde{m}^t)$ is a predictor sequence, typically $\tilde{m}^{t+1} = m^t$. Note that setting $\tilde{m}^{t+1} = 0$ in (OFTRL) results in the celebrated FTRL algorithm.

Using SCNE as a regularizer ($\Phi = \Phi_{\text{ent}}$) as defined in (SCNE), we obtain our algorithm. The algorithm updates a sequence of weights $(w^t)$ accumulating the payoff vectors with an additional optimistic term which is given by the adaptive predictor sequence $(\tilde{m}^t)$. SCNE yields closed-form updates via the EJA exponential and trace normalization to map weights into the generalized simplex.

---

**Optimistic Symmetric Cone Multiplicative Weights Updates**

$$w^{t+1} = \eta \left( \sum_{k=1}^{t} m^k + \tilde{m}^{t+1} \right), \; x^{t+1} = \frac{\exp(w^{t+1})}{\text{tr}(\exp(w^{t+1}))}, \qquad \text{(OSCMWU)}$$

for all $t \geq 1$, where $(\tilde{m}^t)$ is a predictor sequence, typically $\tilde{m}^{t+1} = m^t$.

---

The particular case where $\tilde{m}^{t+1} = 0$ corresponds to the SCMWU algorithm studied in Canyakmaz et al. (2023). When $\mathcal{K} = \mathbb{S}_+^n$, variants of our optimistic algorithm with matrix updates have been recently introduced and analyzed in Vasconcelos et al. (2023) for quantum zero-sum games.

**Remark 2.** *While the terminology of 'multiplicative weights' is clearly justified for the standard simplex, it is less relevant in our more general setting because the exponential of a sum of EJA elements is not equal to the product of the exponentials in general. Therefore the exponential in the update of the decision variable $x^{t+1}$ cannot be split. Nevertheless, we stick to this terminology as (a) it has already been used for the particular case of the matrix multiplicative weight algorithm and (b) it highlights the link between our general algorithm and its popular particular case instances.*

**Remark 3.** *(Exponential computation) Depending on the cone, the exponential map can be made explicit in closed form and computed (or approximated). For instance, this computation is straightforward for the case of the nonnegative orthant and the second-order cone whereas it is a more expensive operation for the PSD cone. We provide a detailed discussion in Appendix B.4. In the PSD case, our algorithm could also*

---

[5]We adopt here the (concave) utility maximization convention like in games rather than the (convex) loss one in OCO.

*be enhanced by using randomization techniques used for matrix multiplicative weights algorithms for solving large semidefinite programs (Baes et al., 2013; Carmon et al., 2019; Yurtsever et al., 2021), we leave this for future work.*

**Proposition 4.** *For any symmetric cone $\mathcal{K}$, the iterates of* (OSCMWU) *coincide with the iterates of* (OFTRL) *with the symmetric cone negative entropy regularizer* $(\Phi = \Phi_{ent})$.

## 5 Regret Analysis and Average Iterate Convergence Guarantees

### 5.1 Individual Regret Bound and Strong Convexity of Symmetric Cone Negative Entropy

The iterates of (OFTRL) satisfy the so-called regret bounded by variation in utilities (RVU) property introduced in Syrgkanis et al. (2015).

**Proposition 5.** *The sequence $(x^t)$ generated by* (OFTRL) *with $\tilde{m}^{t+1} = m^t \in \Delta_{\mathcal{K}}$ for all $t$ and a regularizer $\Phi$ that is 1-strongly convex w.r.t. a norm $\|\cdot\|$ satisfies for any $T \geq 1$, for any $x \in \Delta_{\mathcal{K}}$,*

$$\sum_{t=1}^T f^t(x) - f^t(x^t) \leq \frac{R}{\eta} + \eta \sum_{t=1}^T \|m^t - m^{t-1}\|_*^2 - \frac{1}{4\eta} \sum_{t=1}^T \|x^t - x^{t-1}\|^2,$$

*where $R = \sup_{x \in \Delta_{\mathcal{K}}} \Phi(x) - \inf_{x \in \Delta_{\mathcal{K}}} \Phi(x)$, $\|\cdot\|_*$ is the dual norm and $\langle \cdot, \cdot \rangle$ the EJA inner product.*

**Remark 6.** *The constant $R$ in the regret bound of Proposition 5 implicitly captures the performance dependence on the complexity of the decision space $\Delta_{\mathcal{K}}$. When the regularizer $\Phi$ is the negative symmetric cone entropy $(\Phi = \Phi_{ent})$, note that $R \leq \ln r$ where $r$ is the rank of the EJA. In particular, when $\mathcal{K} = \mathbb{R}_+^n$ or $\mathbb{S}_+^n$ or $\mathbb{H}_+^n$, $r = n$. When $\mathcal{K} = \mathbb{L}_+^n$ (second-order cone), $r = 2$. Notably, the dependence of the regret on the complexity of the decision space is logarithmic.*

**Application to** (OSCMWU)**.** Our goal now is to obtain the regret bound of Proposition 5 for our (OSCMWU) algorithm. Recall now from Proposition 4 that (OSCMWU) can be seen as an (OFTRL) algorithm using the SCNE regularizer $\Phi_{ent}$. Therefore, our main challenge to obtain our regret guarantee is to establish the strong convexity of the SCNE w.r.t. the trace-1 norm. This is one of our key contributions which requires many technical developments relying heavily on the theory of EJAs. The symmetric cone negative entropy is 1-strongly convex w.r.t. the trace-1 norm on the interior of the generalized simplex.

**Theorem 7.** *(Strong convexity of the symmetric cone negative entropy). Let $(\mathcal{J}, \circ)$ be an EJA and let $\mathcal{K}$ be its cone of squares. Then for all $x, y \in int(\Delta_{\mathcal{K}})$, $D_{\Phi_{ent}}(x, y) \geq \frac{1}{2}\|x - y\|_{tr,1}^2$, where we recall that $\Phi_{ent}$ is the symmetric cone negative entropy, i.e. $\Phi_{ent}(x) = tr(x \circ \ln x)$ for all $x \in int(\mathcal{K})$, and $D_{\Phi_{ent}}$ is the Bregman divergence $D_{\Phi_{ent}}(x, y) = tr(x \circ \ln x - x \circ \ln y + y - x)$.[6]*

We contribute a new proof which relies on using a data processing inequality for the specific case of a diagonal mapping[7] in general EJAs followed by an application of Pinsker's inequality (see Eq. (19) in Appendix C.3 for the core step of the proof). We prove in particular that the diagonal mapping can be written as a convex combination of EJA automorphisms and we combine this result with the known joint convexity of the relative entropy for EJAs (Faybusovich, 2016) to obtain the desired inequality. We defer the complete proof to Appendix C.3. An alternative proof for strong convexity can rely on the celebrated duality between strong convexity and strong smoothness (w.r.t. arbitrary norms) which is used to lower bound the Hessian of the negative entropy, see corollary 6.4.5 p.192 in Baes (2006) leading to the same strong convexity constant (equal to 1) as in Theorem 7.

**Remark 8.** *Theorem 7 recovers as special cases the strong convexity of the negative entropy w.r.t. the one norm for the probability simplex and the trace one slice of PSD cones in the optimization (see e.g. Proposition 5.1 in Beck & Teboulle (2003)) and (quantum) information theory literatures (see e.g. Yu (2013)).*

**Remark 9.** *Chen & Pan (2010, Lemma 3.2.c) shows that the SCNE is strictly convex on the symmetric cone (see also Canyakmaz et al. (2023, Lemma 2.1 (ii))). We show that the SCNE is strongly convex with respect to the trace 1 norm, which is a stronger result.*

---

[6]See appendix for more details on the Bregman divergence.
[7]This is the mapping associating to a symmetric matrix its diagonal in the EJA of symmetric matrices.

## 5.2 Sum of Regrets Bound in Symmetric Cone Games

In this section we leverage our strong convexity result for the SCNE (Theorem 7) to provide regret guarantees in SCGs. We start by deriving a constant bound on the sum of regrets of players using (OSCMWU) and the OFTRL framework. Then, we focus on two-player zero-sum SCGs to solve our min-max problem (1) and find an approximate saddle point using the regret guarantees.

If each player runs (OSCMWU), then the sum of regrets of all players is bounded by a constant. The analysis relies on using the RVU bound in Proposition 5. Using Assumption 1, we follow the analysis of Syrgkanis et al. (2015) which is valid for any (OFTRL) algorithm with a strongly convex regularizer w.r.t. any norm (when all players run the same algorithm) to obtain the following result.

**Theorem 10.** *Let Assumption 1 hold and let each player $i \in \mathcal{N}$ runs (OSCMWU) for $T$ rounds on their strategy space $\Delta_{\mathcal{K}_i}$ [8] with a fixed positive stepsize $\eta = 1/(2\sqrt{N \sum_{i=1}^N L_i^2})$ and set $\|\cdot\| = \|\cdot\|_{\mathrm{tr},1}$. Then the sum of regrets is bounded as: $\sum_{i=1}^N r_i(T) \leq 2 \left(\sum_{i=1}^N R_i\right) \cdot \sqrt{N \sum_{i=1}^N L_i^2}$, where $r_i(T)$ is the $i$-th player's individual regret defined in (3) and $R_i = \sup_{x \in \Delta_{\mathcal{K}_i}} \Phi(x) - \inf_{x \in \Delta_{\mathcal{K}_i}} \Phi(x)$.*

## 5.3 Application to Two-Player Zero-Sum Symmetric Cone Games

We consider now the min-max problem:

$$\min_{x \in \Delta_{\mathcal{K}_1}} \max_{y \in \Delta_{\mathcal{K}_2}} f(x, y), \tag{10}$$

where $f : \mathcal{J}_1 \times \mathcal{J}_2 \to \mathbb{R}$ is convex-concave and differentiable, $\mathcal{K}_1, \mathcal{K}_2$ are arbitrary symmetric cones and $\Delta_{\mathcal{K}_1}, \Delta_{\mathcal{K}_2}$ their associated generalized simplices.

We employ well-known results in the online learning literature to obtain an approximate saddle point for the min-max problem using two uncoupled online learning players (see e.g. section 5 in Freund & Schapire (1999), sections 7.1-3 in Cesa-Bianchi & Lugosi (2006) and sections 11.1-2 in Orabona (2019)) and the regret guarantees of Theorem 10. Specifically, we use the link between the sum of individual players' regrets, the so-called duality gap in min-max problems, and the suboptimality gaps for each one of the dual min-max and max-min problems. For the convenience of the reader, we record the result we use in Theorem 20 in Appendix D.2 and provide a complete proof. Note that this result (Theorem 20) is independent from any specific algorithm used and applies to generalized simplices.

We combine this result (Theorem 20) with our regret guarantees obtained in Theorem 10 for (OSCMWU) to solve our saddle point problem (10) of interest for which the generalized min-max theorem holds. To state our result, we respectively define the duality gap $d(\bar{x}, \bar{y})$ and the suboptimality gaps $d_1(\bar{x}), d_2(\bar{y})$ for the min and max dual problems for any $(\bar{x}, \bar{y}) \in \Delta_{\mathcal{K}_1} \times \Delta_{\mathcal{K}_2}$ as follows:

$$d(\bar{x}, \bar{y}) := \max_{y \in \Delta_{\mathcal{K}_2}} f(\bar{x}, y) - \min_{x \in \Delta_{\mathcal{K}_1}} f(x, \bar{y}), \tag{11}$$

$$d_1(\bar{x}) := \max_{y \in \Delta_{\mathcal{K}_2}} f(\bar{x}, y) - v^*, \quad d_2(\bar{y}) := v^* - \min_{x \in \Delta_{\mathcal{K}_1}} f(x, \bar{y}), \tag{12}$$

where $v^*$ is the min-max value of the zero-sum game, i.e. $v^* := \min_{x \in \Delta_{\mathcal{K}_1}} \max_{y \in \Delta_{\mathcal{K}_2}} f(x, y) = \max_{y \in \Delta_{\mathcal{K}_2}} \min_{x \in \Delta_{\mathcal{K}_1}} f(x, y)$.

**Theorem 11.** *Consider a two-player symmetric cone game setting $(N = 2)$ and assume that the game is zero-sum $(u_2 = -u_1 = f)$. Let Assumption 1 hold and suppose that both players run (OSCMWU) with stepsize $\eta = 1/(2\sqrt{2(L_1^2 + L_2^2)})$ for $T \geq \frac{2(\ln r_1 + \ln r_2)\sqrt{2(L_1^2 + L_2^2)}}{\varepsilon}$ rounds where $\varepsilon > 0$ is any desired accuracy and $r_i = \mathrm{rank}(\mathcal{J}_i), i = 1, 2$. Then the average iterate sequences $(\bar{x}_T), (\bar{y}_T)$ defined by $\bar{x}_T = \frac{1}{T} \sum_{t=1}^T x^t, \bar{y}_T = \frac{1}{T} \sum_{t=1}^T y^t$ satisfy for $T \geq 1$, $0 \leq d(\bar{x}_T, \bar{y}_T) \leq \varepsilon, 0 \leq d_1(\bar{x}_T) \leq \varepsilon$ and $0 \leq d_2(\bar{y}_T) \leq \varepsilon$. Moreover $(\bar{x}_T, \bar{y}_T)$ is an $\epsilon$-saddle point of the min-max problem (10), i.e. for all $(x, y) \in \Delta_{\mathcal{K}_1} \times \Delta_{\mathcal{K}_2}, f(\bar{x}_T, y) - \varepsilon \leq f(\bar{x}_T, \bar{y}_T) \leq f(x, \bar{y}_T) + \varepsilon$.*

---

[8]They need not be the same for all players, see the consequence on the regret with the dependence on $R_i$.

The dependence on the complexity of the strategy spaces is logarithmic in the rank of the underlying EJAs (see remark 6). Theorem 11 improves over the $\mathcal{O}(1/\varepsilon^2)$ iteration complexity that can be achieved using the SCMWU algorithm proposed by Canyakmaz et al. (2023). Our analysis deviates from the SCMWU regret analysis developed in Canyakmaz et al. (2023). In particular, we leverage our general SCG setting, rely on the OFTRL framework and the strong convexity of SCNE to establish our result. In the special case where $\mathcal{K}_1 = \mathcal{K}_2 = \mathbb{H}_+^n$, a similar improvement has been established for quantum zero-sum games in Vasconcelos et al. (2023) using a different optimistic mirror-prox like analysis (Nemirovski, 2004) via a variational inequality perspective and several variants of optimistic matrix updates.

## 6 Applications to Min-Max Problems: Metric Learning and Facility Location

In this section, we discuss how to use our unified methodology, our optimistic online algorithm (OSCMWU) and its resulting iteration complexity in the two special min-max applications discussed earlier. Unlike prior work that designs domain-specific solvers, we use the same (OSCMWU) algorithm in all cases. To our knowledge, OSCMWU is the first optimistic algorithm applicable across all symmetric cones in SCGs.

### 6.1 Simplex-Spectraplex Games (e.g. Metric Learning)

**Iteration complexity of** (OSCMWU)**.** In the game (6), given that $x, Y$ are played, the players respectively observe the gain vectors $m_1(x, Y) = -\nabla_x f(x, Y) = -\mathcal{A}^*(Y) - b = -(\langle Y, A_i \rangle)_i - b$, and $m_2(x, Y) = \nabla_Y f(x, Y) = \mathcal{A}(x) + C = \sum_i x_i A_i + C$ where here $\mathcal{A}^*$ denotes the adjoint operator of $\mathcal{A}$. We have $L_1 = L_2 = \max_i \|A_i\|_{\mathrm{tr},\infty}$ and $R_1 = \ln m$, $R_2 = \ln d$. Thus, by Theorem 11, to obtain an $\epsilon$-saddle point of the min-max problem it suffices for both players to run (OSCMWU) with stepsize $\eta = 1/(2 \max_i \|A_i\|_{\mathrm{tr},\infty})$ for $T \geq \frac{4(\ln m + \ln d) \max_i \|A_i\|_{\mathrm{tr},\infty}}{\epsilon}$ iterations.

**Simulations.** We consider an instance of the metric learning application (Application 1 in section 3.2) using the Iris dataset (Fisher, 1936) ($d = 4$) loaded via scikit-learn (Pedregosa et al., 2011). We standardize the features, sample $|\mathcal{S}| = 400$ similar (same-label) pairs and $|\mathcal{D}| = 400$ dissimilar (different-label) pairs. We then solve the resulting simplex–spectraplex game (5) by running SCMWU and its optimistic variant (OSCMWU) for $T = 9000$ iterations with stepsizes $\eta_x = \eta_Y = 20$ starting from a random initialization. We report the duality gap of the average iterates as defined in (11) on a log scale. The plotted curves in Figure 1 correspond to the average over 5 independent seeds (which resample $\mathcal{S}, \mathcal{D}$ and the initialization) and the shaded area is $\pm$ one standard deviation. Figure 1 shows a slight advantage for (OSCMWU) compared to its non-optimistic counterpart SCMWU.

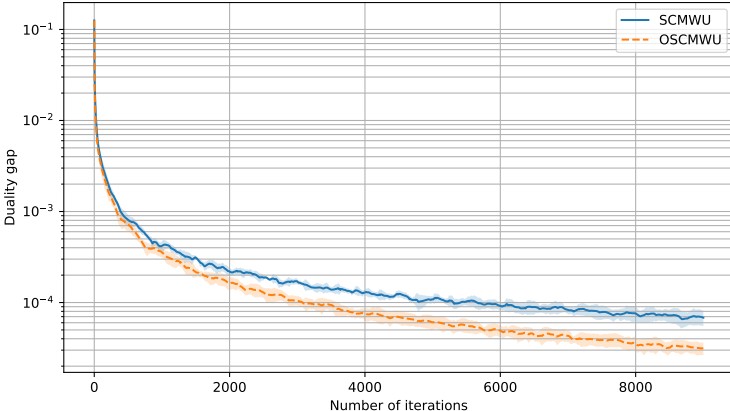

Figure 1: Duality gap of the average iterates versus number of iterations for OSCMWU and SCMWU for a distance metric-learning application on the Iris dataset (mean $\pm$ std over 5 seeds).

**Prior work.** In contrast to prior work, we use a single algorithm (OSCMWU) for all our applications of SCGs. Existing literature proposes algorithms which are tailored to specific applications. Problem (5) can

be solved using the smoothing technique (Nesterov, 2007) in $\mathcal{O}(1/\epsilon)$ iterations and a square root logarithmic dependence on the dimension $d$. Our iteration complexity is similar. Both methods require a full eigenvalue decomposition per iteration. This per-iteration cost can be significantly reduced using techniques such as rank sketches (Baes et al., 2013; Carmon et al., 2019) while keeping a similar total number of iterations. Ying & Li (2012) proposed a Frank-Wolfe method using the smoothing technique which only requires to compute the largest eigenvector of a matrix at each iteration which costs $\mathcal{O}(d^2)$ runtime and improves over the per-iteration cost of a full eigenvalue decomposition. However, their algorithm requires $\mathcal{O}(1/\epsilon^2)$ iterations. We also refer the reader to Garber & Hazan (2016) for further improvements for solving instances of the general problem (6) with approximation algorithms running in sublinear time in the size of the data. The problem can also be solved very efficiently and with high precision using interior-point methods but their per-iteration cost becomes prohibitive for large scale problems for which first-order methods are preferred.

### 6.2   Second-Order Cone Min-Max Games (e.g. Fermat-Weber Problem)

**Iteration complexity of** (OSCMWU). For this problem, it can be immediately seen that $m_1(x, y) = -\nabla_{\bar{x}} f(\bar{x}, y) = \tilde{A}^T y, m_2(x, y) = \nabla_y f(\bar{x}, y) = \tilde{A}\bar{x} - \bar{b}$. It follows that the Lipschitz constants satisfy $L_1 = L_2 = \max_i \|A_i\|_{\mathrm{tr}, \infty}$ and $R_1 = \ln\big(\mathrm{rank}(\mathbb{L}^{d+1})\big) = \ln 2, R_2 = p \ln\big(\mathrm{rank}(\mathbb{L}^{d+1})\big) = p \ln 2$. By Theorem 11, in order to obtain an $\epsilon$-saddle point it suffices for both players to run (OSCMWU) with stepsize $\eta = \frac{1}{2L}$ for $T \geq \frac{4(p+1)L \ln 2}{\epsilon}$ iterations.

**Simulations.** We consider a synthetic instance of the Fermat-Weber (sum-of-norms) problem as described in application 1 in section 3.2, in dimension $d = 20$ with $p = 50$ target points and constraint radius $R = 5$. For each seed, we generate an instance $B = \{b_i\}_{i=1}^p \subset \mathbb{R}^d$ by sampling i.i.d. Gaussian directions, normalizing each sample, and then scaling each $b_i$ to have a random norm uniformly distributed in $[0.5R, 1.5R]$. We solve the associated second-order-cone min-max game (9) by running SCMWU and its optimistic variant (OSCMWU) for $T = 10000$ iterations with constant stepsizes $\eta_x = \eta_y = 7 \times 10^{-2}$. In Figure 2, we report (i) the primal objective $g(\bar{x}_t)$ as defined in (7) evaluated at the average iterate $\bar{x}_t$, and (ii) the duality gap computed at the averaged iterates $(\bar{x}_t, \bar{y}_t)$ as in (11), plotted on a logarithmic scale. The curves correspond to the mean over 5 seeds, and the shaded region indicates $\pm$ one standard deviation. The resulting two-panel plot shows that the objective function decreases and stabilizes as expected and the duality gap vanishes as shown in Theorem 11 with an advantage for (OSCMWU) compared to SCMWU.

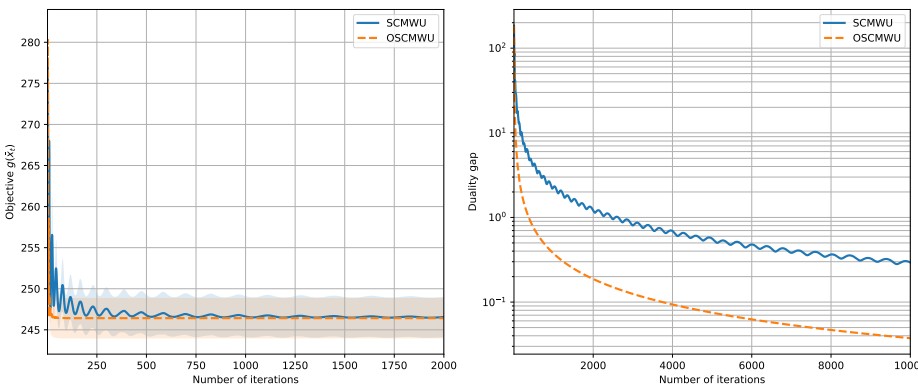

Figure 2: (Left) Objective function values of the average iterates (Right) Duality gap of the average iterates for OSCMWU and SCMWU for a facility location problem (mean $\pm$ std over 5 seeds).

**Online facility location variant.** We now consider an online variant of the facility location problem introduced in section 3.2. Consider a sequential setting in which demands arrive over time (streaming demand) and the decision maker must update the facility location on the fly. At each round $t = 1, 2, \ldots, T$, a new point $b_t \in \mathbb{R}^d$ and a linear map $A_t \in \mathbb{R}^{m \times d}$ are revealed after the learner has committed to a location $x_t \in C := \{x \in \mathbb{R}^d : \|x\|_2 \leq R\}$. The instantaneous incurred loss is $g_t(x_t) := \|A_t x_t - b_t\|_2$. Similarly to section 3.2, we define an *online* saddle-point loss $f_t(\bar{x}, y) := \langle \tilde{A}_t \bar{x} - \bar{b}_t, y \rangle$ for any $\bar{x} = (1/2, x/(2R)) \in$

$\Delta_{\mathbb{L}_+^{d+1}}$, $y \in \Delta_{\mathbb{L}_+^{d+1}}$, where $\tilde{A}_t := 2R(0 \quad \bar{A}_t)$, $\bar{A}_t = (0, A_t^\top)^\top$ and $\bar{b}_t := (0, b_t)$. Running (OSCMWU) online corresponds to updating $\bar{x}_t$ using the first-order feedback $-\nabla_{\bar{x}} f_t(\bar{x}_t, y_t) = \tilde{A}_t^\top y_t$ and updating $y_t$ using $\nabla_y f_t(\bar{x}_t, y_t) = \tilde{A}_t \bar{x}_t - \bar{b}_t$. In the experiment of Figure 3, we consider a setting with $d = m = 10$, radius $R = 1$, and horizon $T = 20000$. The data stream $(A_t, b_t)$ is generated from a temporally correlated (predictable) process: $A_t \equiv A$ is sampled once with i.i.d. standard Gaussian entries and then rescaled so that $\|A\|_2 = 1$, while $b_t \in \mathbb{R}^d$ follows an AR(1) recursion $b_t = \rho b_{t-1} + \sqrt{1 - \rho^2}\, \xi_t$ with $\xi_t \sim \mathcal{N}(0, \sigma^2 I)$ (here $\rho = 0.6$, $\sigma = 0.12$), followed by a normalization enforcing $\|b_t\|_2 = 1$. We run SCMWU and (OSCMWU) with the same constant stepsizes $\eta_x = \eta_y = 10$ from the interior initialization $(1/2, 0, \ldots, 0) \in \Delta_{\mathbb{L}_+^{d+1}}$. We report the time-scaled sum of regrets $(r_1(t) + r_2(t))/t$ (mean $\pm$ one standard deviation over 5 independent seeds, which resample the entire stream), as defined and bounded in Theorem 10. Figure 3 shows that the time-scaled sum of regrets vanishes for both (OSCMWU) and SCMWU, with an advantage for OSCMWU in this predictable online learning setting.

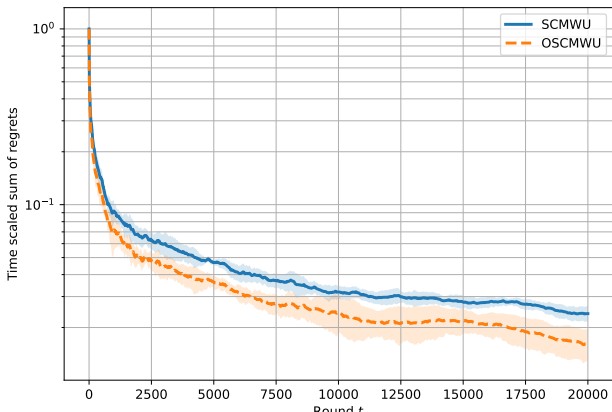

Figure 3: Time-scaled sum of regrets for OSCMWU and SCMWU for an online facility location problem (mean $\pm$ std over 5 seeds).

**Prior work.** The sum-of-norms problem can be solved using an extension of the smoothing technique of Nesterov to EJAs studied in Baes (2006) (section 6.5 p.194) with a number of iterations of the order $\mathcal{O}(1/\epsilon)$ and a $\mathcal{O}(mdp)$ per-iteration cost. (OSCMWU) enjoys similar guarantees (see the straightforward computation of the exponential for the second-order cone case in Appendix B.4). For comparison, the interior point method of Xue & Ye (1997) requires fewer iterations $\mathcal{O}(\sqrt{p}(\log(\max_j \|b_j\|/\epsilon) + \log p))$ as a function of the desired accuracy $\epsilon$ but requires a much higher per-iteration cost $\mathcal{O}(m^3 + pm^2 n)$. We note that none of the previous methods is online. In contrast to our method, they are offline methods tailored for this particular problem.

For more general hybrid SC min-max games with product of cones of different types (SOC, SDP), see appendix E.

## 7 Conclusion

We introduced the class of Symmetric Cone Games (SCGs) which unifies a variety of multi-player games with structured strategy spaces, including normal-form, quantum and geometric settings. SCGs provide a principled foundation for structured game-theoretic learning, bridging spectral and geometric optimization in a unified framework. We proposed OSCMWU, a universal optimistic online learning algorithm with closed-form updates, and established convergence guarantees for equilibrium computation in two-player zero-sum SCGs. Our framework paves the way for more unified approaches to structured game-theoretic learning exploiting the structure of strategy spaces beyond simplices or generic convex sets. Promising directions for future work include extending beyond symmetric cones, tackling non-zero-sum games, and developing scalable implementations for high-dimensional and low-rank settings for the case of the PSD cone.

## Acknowledgments

We thank the anonymous reviewers for their valuable feedback, which helped improve the manuscript.

This work is supported by the MOE Tier 2 Grant (MOE-T2EP20223-0018), Ministry of Education Singapore (SRG ESD 2024 174), the CQT++ Core Research Funding Grant (SUTD) (RS-NRCQT-00002), and partially by Project MIS 5154714 of the National Recovery and Resilience Plan, Greece 2.0, funded by the European Union under the NextGenerationEU Program.

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

## Contents

## A   Related Work

**Multiplicative weight updates.** The Multiplicative Weights Update (MWU) algorithm has a long history percolating several fields including game theory, online learning, optimization, machine learning and computer science. We refer the reader to Arora et al. (2012) for a nice survey. Beyond its classical form over the probability simplex, extensions to more complex structures—such as the Matrix Multiplicative Weights Update (MMWU) for density matrices (Tsuda et al., 2005; Warmuth & Kuzmin, 2006; 2012; Arora & Kale, 2016)—have broadened its applicability to semi-definite programming (Arora et al., 2005; Arora & Kale, 2016), quantum computing (Jain et al., 2011) and graph sparsification (Allen-Zhu et al., 2015). Carmon et al. (2019) proposed rank-one sketch methods for the matrix multiplicative weight method to reduce the expensive computational cost of a full matrix exponentiation while preserving the MMWU regret guarantees. We refer the reader to the discussion therein for more details regarding this line of work. Canyakmaz et al. (2023) introduced the symmetric cone multiplicative weights update algorithm (SCMWU) for online learning over the trace-one slice of a symmetric cone. (SCMWU) unifies the different versions of multiplicative weights updates algorithms mentioned above. Our (OSCMWU) algorithm enhances (SCMWU) with an additive optimistic term in the weights sequence. Zhao (2023) analyzed the multiplicative gradient method for an optimization problem in which the objective function is neither Lipschitz nor smooth.

**Optimistic online learning.** A key performance measure in online learning and game theory is regret. For min-max problems for which the min-max theorem holds, it is well-known that if the average regret is bounded above by $\epsilon$, then the time-averaged iterate $(\bar{x}, \bar{y})$ is an $\epsilon$-saddle point. In worst-case contexts in which the environment is modeled in a fully adversarial way, average regret typically vanishes at the rate of $\mathcal{O}(T^{-1/2})$, a bound known to be tight. However, this worst-case analysis is too pessimistic when players adopt predictable algorithms in games. In these settings, optimistic learning algorithms have achieved significant improvements, reducing average regret rates to $\mathcal{O}(T^{-1})$ in both two-player zero-sum games and multi-player normal-form games (Daskalakis et al., 2011; Chiang et al., 2012; Rakhlin & Sridharan, 2013b; Syrgkanis et al., 2015; Daskalakis et al., 2021). These impactful advances have been established for games where strategies lie in the standard simplex (Daskalakis et al., 2021; Piliouras et al., 2022) and extended to general convex subsets of the Euclidean space (Syrgkanis et al., 2015; Farina et al., 2022a) and polyhedral convex games (Farina et al., 2022b). Recently, Vasconcelos et al. (2023) showed that improved rates can also be achieved in the case of zero-sum quantum games using several different optimistic matrix multiplicative weight (OMMWU) algorithms, some of which can be seen as a particular case of our (OSCMWU) algorithm instantiated with the PSD cone. Our analysis in this work is different as it relies on the OFTRL framework rather than the optimistic mirror prox viewpoint and holds for arbitrary symmetric cones.

**Symmetric cone programming using SCMWU.** Recent works (Zheng et al., 2024; Zheng & Tan, 2024) used the Symmetric Cone Multiplicative Weights Update (SCMWU) to solve specific Symmetric Cone Programs (SCPs) via binary search, where in each step of the binary search a min-max problem is solved approximately in order to solve the approximate feasibility problem (of the intersection of the feasible set and the sublevel set of the objective function). In the specific problems tackled by both of those works, the strategy space of the min-max game is the trace-one slice of a symmetric cone on one side (where the strategy is updated using SCMWU) and a convex set on the other (where the strategy is played by an oracle). Thus, while their method for solving SCPs uses SCMWU in a game-playing framework, it is not

easily amenable to the speedup that optimistic methods enjoy due to there being no stability guarantees for the iterates in the convex set (played by an oracle). They are, however, able to handle different sets of problems since additional constraints (that are not just trace constraints) can be handled by the oracle. Thus, while their work is related, we regard their method and applications as parallel to those we present in this work.

**Applications of symmetric cone games.** Some of our applications have been extensively studied in the semidefinite programming (SDP) and second-order cone programming (SOCP) literatures. For instance, problem (6) has been studied in the SDP literature with a special emphasis on designing efficient first-order methods for large scale problems using different randomization techniques (Baes et al., 2013; Yurtsever et al., 2021; d'Aspremont & El Karoui, 2014; Carmon et al., 2019). Sum-of-norms problems have been discussed for instance in Alizadeh & Goldfarb (2003) (see section 2.2). Other norm minimization problems such as maximum norm minimization also fit our symmetric cone game setting and we can use (OSCMWU) similarly. Nevertheless, our symmetric cone game setting captures arbitrary cone examples beyond the special cases of the simplex-spectraplex or second-order cone games we consider for illustrating the expressiveness of the class of symmetric cone games. Besides the applications we consider, Jambulapati & Tian (2024) have considered box-simplex games as well as box-spectraplex games and proposed faster first-order algorithms relying on area convexity (Sherman, 2017).

# B    Background on Symmetric Cones and EJAs

## B.1    Definition of symmetric cones

**Definition 12.** *(Symmetric cone) Let $(V, \psi(\cdot, \cdot))$ be an inner product space. A cone $\mathcal{K} \subset V$ is symmetric if:*

- *$\mathcal{K}$ is self-dual, i.e. $\mathcal{K}^* = \mathcal{K}$ where $\mathcal{K}^* := \{y \in V : \psi(y, x) \geq 0, \forall x \in \mathcal{K}\}$ is the dual cone.*

- *$\mathcal{K}$ is homogeneous, i.e. for any $u, v \in int(\mathcal{K})$, there exists an invertible linear map $T$ s.t. $T(u) = v$ and $T(\mathcal{K}) = \mathcal{K}$.*

## B.2    Rank of an EJA

The minimal polynomial of an element $x \in \mathcal{J}$ is the monic polynomial $p \in \mathbb{R}[x]$ of minimum degree for which $p(x) = 0$, and the degree of an element $x \in \mathcal{J}$ is the degree of its minimal polynomial. The rank of a Jordan algebra is the maximum degree of its elements.

## B.3    Examples of symmetric cones and their corresponding EJAs

- **The nonnegative orthant $\mathbb{R}^n_+$.** The corresponding EJA is $\mathbb{R}^n$ equipped with Jordan product $x \circ y = (x_1 y_1, \ldots, x_n y_n)$, and the identity element is the all-ones vector $e = 1_n$. The rank of this EJA is $n$, and the spectral decomposition of $x \in \mathbb{R}^n$ is $x = \sum_i x_i e_i$ where $e_i$ is the standard $i$-th basis vector. The trace of $x \in \mathbb{R}^n$ is then the sum of its components, i.e., $\mathrm{tr}(x) = \sum_i x_i$, and the canonical EJA inner product is the Euclidean inner product $\langle x, y \rangle = \mathrm{tr}(x \circ y) = \sum_i x_i y_i$.

- **The second-order cone $\mathbb{L}^n_+$ (also known as the Lorentz cone).** This is the set $\{x = (x_1, \bar{x}) \in \mathbb{R} \times \mathbb{R}^{n-1} : \|\bar{x}\|_2 \leq x_1\}$, and its corresponding EJA is $\mathbb{L}^d$, i.e., the set $\mathbb{R}^n$ equipped with the Jordan product $x \circ y = (\bar{x}^\top \bar{y}, x_1 \bar{y} + y_1 \bar{x})$ where $x = (x_1, \bar{x}), y = (y_1, \bar{y}) \in \mathbb{R} \times \mathbb{R}^{n-1}$ and we use $\bar{x}^\top \bar{y}$ to denote $\sum_{i=1}^{n-1} \bar{x}_i \bar{y}_i$. The identity element is $e = (1, 0_{n-1})$. The rank of this EJA is 2, and the spectral decomposition of a point $(s, x) \in \mathbb{L}^n$ is

$$(s, x) = (s + \|x\|_2) \, q_+ + (s - \|x\|_2) \, q_-, \tag{13}$$

where $q_\pm = \frac{1}{2}\left(1, \pm \frac{x}{\|x\|_2}\right)$. The trace is given by $\mathrm{tr}((s, x)) = 2s$ and the canonical EJA inner product is twice the Euclidean inner product, i.e.,

$$\langle x, y \rangle = 2x^\top y.$$

(These are the conventions we use in this work; it should be noted that some sources choose to scale the Jordan product by a factor of $\frac{1}{\sqrt{2}}$ to remove the factor of 2 from the expressions of the trace and the EJA inner product.)

- **The PSD cone** $\mathbb{S}_+^n$, i.e., the set of $n \times n$ symmetric positive semidefinite matrices with real entries. The corresponding EJA is $\mathbb{S}^n$, the set of $n \times n$ symmetric matrices with real entries, equipped with the Jordan product

$$X \circ Y = \frac{XY + YX}{2}, \tag{14}$$

  i.e. the symmetrized matrix product. The identity element is the identity matrix $e = I_n$. The rank of this EJA is $n$, the spectral decomposition is the usual spectral decomposition for symmetric matrices, and the trace is the usual matrix trace. The EJA inner product is the Hilbert-Schmidt inner product $\langle X, Y \rangle = \mathrm{tr}(X \circ Y) = \mathrm{tr}(XY)$.

- **The set** $\mathbb{H}_+^n$ **of** $n \times n$ **Hermitian PSD matrices with complex entries.** The corresponding EJA is the set of $n \times n$ Hermitian matrices with complex entries, equipped with the Jordan product (14). The relevant properties are the same as for $\mathbb{S}_+^n$ above.

- **Product cones.** Any Cartesian product of the above symmetric cones is also a symmetric cone, and the corresponding EJA is the product of their corresponding EJAs. The identity element of this EJA is the product of identity elements, and the spectral decomposition of an element is the product of the spectral decompositions of its components. The rank/trace of an element is simply the sum of the ranks/traces of its components.

Table 2: Examples of Euclidean Jordan Algebras and the corresponding Symmetric Cones

| EJA $\mathcal{J}$ | | Inner product $\langle x, y \rangle$ | Jordan product $x \circ y$ | Cone of squares $\mathcal{K}$ | |
|---|---|---|---|---|---|
| Euclidean space | $(\mathbb{R}^n)$ | $\sum_{i=1}^n x_i y_i$ | $(x_i y_i)_{i=1,\cdots,n}$ [#] | nonnegative orthant | $(\mathbb{R}_+^n)$ |
| Real sym. matrices | $(\mathbb{S}^n)$ | $\mathrm{tr}(xy)$ | $\frac{1}{2}(xy + yx)$ [†] | PSD cone[1] | $(\mathbb{S}_+^n)$ |
| Jordan spin algebra[2] | $(\mathbb{L}^n)$ | $2\sum_{i=1}^n x_i y_i$ | $(\bar{x}^\top \bar{y}, x_1 \bar{y} + y_1 \bar{x})$ [&] | second-order cone[3] | $(\mathbb{L}_+^n)$ |

[1] This is the set of real symmetric positive semidefinite matrices. [2] This is the set $\mathbb{R} \times \mathbb{R}^{n-1}$. [3] This cone is the set $\mathbb{L}_+^n = \{x = (x_1, \bar{x}) \in \mathbb{R} \times \mathbb{R}^{n-1} : \|\bar{x}\| \le x_1\}$, and is also known as the Lorentz cone. [#] This is the coordinatewise product vector, a.k.a. the Hadamard or Schur product. [†] The product used here is the standard matrix product. [&] Here $x = (x_1, \bar{x}), y = (y_1, \bar{y}) \in \mathbb{R} \times \mathbb{R}^{n-1}$. We use $\bar{x}^\top \bar{y}$ to denote $\sum_{i=1}^{n-1} \bar{x}_i \bar{y}_i$.

### B.4 Exponential computation in (OSCMWU)

In this section, we briefly discuss the computation of the exponential of an element $x \in \mathcal{J}$ where $J$ is an EJA. We provide comments on a case by case basis.

**Euclidean space** $\mathbb{R}^n$**.** In this case, the exponential computation is straightforward and consists in computing the exponential of each one of the coordinates, i.e. for a vector $x \in \mathbb{R}^n$, $\exp(x) = (\exp(x_i))_{i=1,\cdots,n} \in \mathbb{R}^d$.

**Jordan spin algebra** $\mathbb{L}^n$**.** Given the spectral decomposition (13) which is easy to compute in time linear in the dimension $n$, we obtain

$$\exp((s, x)) = \exp\left(s + \|x\|_2\right) q_+ + \exp\left(s - \|x\|_2\right) q_-. \tag{15}$$

Therefore the computation of the exponential is straightforward.

**Real symmetric matrices** $\mathbb{S}^n$**.** In this case, the computation of the matrix exponential is more expensive and can be performed (approximately) in as much as $\tilde{\mathcal{O}}(n^3)$ time for a $n \times n$ matrix (the notation $\tilde{\mathcal{O}}(\cdot)$ hides polylogarithmic factors in $n$ and in $1/\epsilon$ where $\epsilon$ refers to the accuracy of the solution) by computing an eigenvalue decomposition of the symmetric matrix and exponentiating the eigenvalues. We refer the reader to section 7 in Arora & Kale (2016) for more details and potential improvements for faster matrix exponentiation.

**Product EJAs.** In this more general setting, the exponential can be computed separately for each one of the EJAs.

### B.5 Trace-p norms

**Theorem 13.** *The trace-$\infty$ norm* $\|x\|_{\mathrm{tr},\infty} = \max_j |\lambda_j(x)|$ *is the dual norm of the trace-1 norm* $\|x\|_{\mathrm{tr},1} = \sum_j |\lambda_j|$.

*Proof.* Recall that the dual norm $\|x\|_{\mathrm{tr},1,*} = \sup_{\|v\|_{\mathrm{tr},1}=1} |\langle v, x\rangle|$. We shall prove separately that $\|x\|_{\mathrm{tr},1,*} \geq \|x\|_{\mathrm{tr},\infty}$ and $\|x\|_{\mathrm{tr},1,*} \leq \|x\|_{\mathrm{tr},\infty}$.

($\geq$) For a given $x \in \mathcal{J}$ with spectral decomposition $\sum_j \lambda_j q_j$, let $k \in \operatorname{argmax}_j |\lambda_j|$ so that $\|x\|_{\mathrm{tr},\infty} = |\lambda_k|$ and set $\tilde{v} := \operatorname{sign}(\lambda_k) q_k$. Since $\tilde{v}, x$ share the Jordan frame $\{q_j\}$, we have

$$\langle \tilde{v}, x\rangle = \operatorname{tr}(\tilde{v} \circ x) = \operatorname{sign}(\lambda_k)\lambda_k = |\lambda_k| = \|x\|_{\mathrm{tr},\infty}.$$

Since $\|\tilde{v}\|_{\mathrm{tr},1} = 1$, we then have $\|x\|_{\mathrm{tr},1,*} = \sup_{\|v\|_{\mathrm{tr},1}=1} |\langle v, x\rangle| \geq \|x\|_{\mathrm{tr},\infty}$.

($\leq$) For a given $x \in \mathcal{J}$ we have $\|x\|_{\mathrm{tr},\infty} = \max_j |\lambda_j(x)|$, thus

$$-\|x\|_{\mathrm{tr},\infty} e \preceq_{\mathcal{K}} x \preceq_{\mathcal{K}} \|x\|_{\mathrm{tr},\infty} e, \tag{16}$$

where we use the generalized inequality $x \preceq_{\mathcal{K}} y$ for $x, y \in \mathcal{J}$ to denote that $y - x \in \mathcal{K}$. Then, by self-duality of the symmetric cone, we can take the inner product of the inequality chain (16) with any $v \in \Delta_{\mathcal{K}}$ to give[9]

$$-\|x\|_{\mathrm{tr},\infty} \leq \langle v, x\rangle \leq \|x\|_{\mathrm{tr},\infty}, \qquad \forall v \in \Delta_{\mathcal{K}},$$

i.e., $\sup_{v \in \Delta_{\mathcal{K}}} |\langle v, x\rangle| \leq \|x\|_{\mathrm{tr},\infty}$. Finally, by Lemma 26 we have

$$\|x\|_{\mathrm{tr},1,*} = \sup_{\|v\|_{\mathrm{tr},1}=1} |\langle v, x\rangle| = \sup_{v \in \Delta_{\mathcal{K}}} |\langle v, x\rangle| \leq \|x\|_{\mathrm{tr},\infty}.$$

$\square$

### B.6 Bregman divergence of the symmetric cone negative entropy

Let $\Phi : C \to \mathbb{R}$ be a continuously differentiable and strictly convex function defined on the convex set $C$. The Bregman divergence $D_\Phi(x, y)$ associated with $\Phi$ for the points $x, y \in C$ is the difference between the value of $\Phi$ at $x$ and the first-order Taylor expansion of $F$ around $y$ evaluated at $y$, i.e., $D_\Phi(x, y) = \Phi(x) - \Phi(y) - \langle \nabla\Phi(y), x - y\rangle$. The Bregman divergence associated to the SCNE $\Phi_{\mathrm{ent}}$ defined above is given by $D_{\Phi_{\mathrm{ent}}}(x, y) = \operatorname{tr}(x \circ \ln x - x \circ \ln y + y - x)$ for every $x, y \in \operatorname{int}(\mathcal{K})$.

## C Proofs for Section 4

### C.1 Proof of Proposition 4

The proof follows similar lines as Lemma 4.1 in Canyakmaz et al. (2023) showing the equivalence between (non-optimistic) SCMWU and FTRL.

---

[9]We use the fact that $\langle v, e\rangle = \operatorname{tr}(v \circ e) = \operatorname{tr}(v) = 1$ for all $v \in \Delta_{\mathcal{K}}$.

## C.2  Proof of Proposition 5

This result echoes Proposition 7 in Syrgkanis et al. (2015) which relies on the combination of Theorem 19 (a regret bound) and Lemma 20 (a stability result) therein. Both these results hold in our setting as the proof follows the exact same lines upon replacing the inner product by the EJA inner product, the pair of dual norms $(\|\cdot\|, \|\cdot\|_*)$ by dual norms defined on the EJA and the simplex $\Delta(\mathbb{S}_i)$ (where $\mathbb{S}_i$ is a finite strategy space for player $i$ in their context) by the generalized simplex $\Delta_{\mathcal{K}}$. We do not reproduce the proof here for conciseness.

## C.3  Proof of Theorem 7: Strong convexity of the SCNE

We provide a detailed proof. The reader familiar with EJAs might directly check Eq. (19) and the proof of the data processing inequality for the diagonal mapping (Theorem 14). Let $\mathcal{K}$ be the cone of squares associated to an EJA $\mathcal{J}$ and denote by $r$ its rank. Let $x, y \in \mathcal{K}$ s.t. $tr(x) = tr(y) = 1$. There exists a Jordan frame $\{q_i\}_{1 \le i \le r}$ and scalars $\{\lambda_i\}_{1 \le i \le r}$ s.t. $x - y = \sum_{i=1}^{r} \lambda_i q_i$. We further suppose without loss of generality that $\|q_i\| = 1$ so that the Jordan frame is an orthonormal basis of a subspace of the EJA $\mathcal{J}$. We will use the canonical inner product provided by the trace throughout this proof.

Consider the following mapping:

$$
\begin{aligned}
T: \quad & \mathcal{J} \to \mathcal{J} \\
& z \mapsto \sum_{i=1}^{r} \langle z, q_i \rangle q_i.
\end{aligned}
\tag{17}
$$

We first observe that $T(int(\mathcal{K})) \subseteq int(\mathcal{K})$ and $tr(T(x)) = tr(T(y)) = 1$. In other words, the mapping $T$ is trace preserving and maps the interior of the cone to a subset of the same cone. In particular, this ensures that the Bregman divergence between the points $T(x)$ and $T(y)$ is well-defined as this will be useful in the rest of the proof. We provide a brief proof of the simple aforementioned facts:

- Let $x \in int(\mathcal{K})$, we prove that $T(x) \in int(\mathcal{K})$. Since $x \in \mathcal{K} = \mathcal{K}^*$ (by definition of a symmetric cone), it follows that for any $z \in \mathcal{K}, \langle x, z \rangle \ge 0$. Moreover, for every $1 \le i \le r, q_i = q_i^2 \in \mathcal{K}$. Therefore $\langle x, q_i \rangle > 0$ and the inequality is strict since $x \in int(\mathcal{K}) = int(\mathcal{K}^*)$.

- Observe that $tr(T(x)) = \sum_{i=1}^{r} \langle x, q_i \rangle tr(q_i) = \sum_{i=1}^{r} \langle x, q_i \rangle = \langle x, \sum_{i=1}^{r} q_i \rangle = \langle x, e \rangle = tr(x) = 1$, where we have used the linearity of the trace, the definition of a Jordan frame and the canonical inner product $(\langle \cdot, \cdot \rangle = tr(\cdot \circ \cdot))$.

We also define the mapping:

$$
\begin{aligned}
u: \quad & \mathcal{J} \to \mathbb{R}^r \\
& z \mapsto (\langle z, q_i \rangle)_{1 \le i \le r}.
\end{aligned}
\tag{18}
$$

We now have the following set of inequalities:

$$
D_{\Phi}(x, y) \underset{(a)}{\ge} D_{\Phi}(T(x), T(y)) \underset{(b)}{=} \mathrm{KL}(u(x) \| u(y)) \underset{(c)}{\ge} \frac{1}{2} \|u(x) - u(y)\|_1^2 \underset{(d)}{=} \frac{1}{2} \|x - y\|_{\mathrm{tr}, 1}^2,
\tag{19}
$$

where (a) follows from the projection inequality (see Theorem 14 below for a statement). (b) stems from writing $D_{\Phi}(T(x), T(y)) = D_{\Phi}(\sum_{i=1}^{r} \langle x, q_i \rangle q_i, \sum_{i=1}^{r} \langle y, q_i \rangle q_i)$ and the spectral definition of the Bregman divergence in terms of the eigenvalues of $x$ and $y$ in the Jordan frame $\{q_i\}$, (c) is an application of Pinsker's inequality in the Euclidean space $\mathbb{R}^r$ using the definition of the mapping $u$, (d) follows from using the definition of the linear mapping $u$ and writing:

$$
\|u(x) - u(y)\|_1 = \|u(x - y)\|_1 = \sum_{i=1}^{r} |\langle x - y, q_i \rangle| = \|T(x - y)\|_{\mathrm{tr}, 1} = \|x - y\|_{\mathrm{tr}, 1},
\tag{20}
$$

where the last inequality follows from observing that

$$T(x - y) = \sum_{i=1}^{r} \langle q_i, x - y \rangle q_i = \sum_{i=1}^{r} \langle q_i, \sum_{j=1}^{r} \lambda_j q_j \rangle q_i = \sum_{i=1}^{r} \lambda_i \langle q_i, q_i \rangle q_i = \sum_{i=1}^{r} \lambda_i q_i = x - y. \tag{21}$$

Note that we have used here the decomposition of $x - y$, the definition of a Jordan frame and the fact that the Jordan frame consists of normalized idempotents. Note also that the above identity $T(x - y) = x - y$ is precisely why we have considered the Jordan frame associated with $x - y$ from the beginning of the proof.

**Theorem 14** (Projection inequality). *Let $J$ be a simple rank-$r$ EJA and let $\mathcal{K}$ be its cone of squares. Let $\{q_i\}_{i=1}^{r}$ be a Jordan frame. Define*

$$T : \quad \mathcal{J} \to \mathcal{J}$$

$$z \mapsto \sum_{i=1}^{r} \langle z, q_i \rangle q_i. \tag{22}$$

*Then for every $x, y \in int(\mathcal{K})$,*

$$D_\Phi(T(x), T(y)) \leq D_\Phi(x, y). \tag{23}$$

*Proof.* The idea of the proof is to write the diagonal operator $T$ as a convex combination of automorphisms of the Jordan algebra and then use the joint convexity of the relative entropy function shown in Corollary 3.4 p. 8 in Faybusovich (2016) to obtain the desired inequality. Before providing the proof for EJAs, we provide first some motivation for the proof based on a proof of the special case of the EJA of real symmetric matrices. The reader familiar with such a proof might skip item (i).

**(i) Motivation and intuition.** Our proof is inspired from some known observations made in Problems II.5.4-II.5.5 in Bhatia (2013). Consider the following $3 \times 3$ block matrices

$$A = \begin{pmatrix} A_{11} & A_{12} & A_{13} \\ A_{21} & A_{22} & A_{23} \\ A_{31} & A_{32} & A_{33} \end{pmatrix}, \quad U_1 = \begin{pmatrix} I & 0 & 0 \\ 0 & I & 0 \\ 0 & 0 & -I \end{pmatrix}, \quad U_2 = \begin{pmatrix} I & 0 & 0 \\ 0 & -I & 0 \\ 0 & 0 & I \end{pmatrix}, \tag{24}$$

where $I$ are identity matrices with dimensions compatible with the blocks of the matrix $A$. It follows from simple computations that

$$\mathcal{C}_3(A) := \frac{1}{2}(A + U_1 A U_1) = \begin{pmatrix} A_{11} & A_{12} & 0 \\ A_{21} & A_{22} & 0 \\ 0 & 0 & A_{33} \end{pmatrix}. \tag{25}$$

Similarly, as if we apply the same treatment to the $2 \times 2$ non-zero block of the matrix $A$,

$$\mathcal{C}_2(A) := \frac{1}{2}(\mathcal{C}_3(A) + U_2 \, \mathcal{C}_3(A) \, U_2) = \begin{pmatrix} A_{11} & 0 & 0 \\ 0 & A_{22} & 0 \\ 0 & 0 & A_{33} \end{pmatrix}. \tag{26}$$

We have obtained the diagonal block of $A$ from the initial matrix $A$ by applying so-called pinching operations. In particular, observe that $U_1^2 = U_2^2 = I$. This implies that $\mathcal{C}_3$ is an average of 2 automorphisms and so is $\mathcal{C}_2$ by composition of automorphisms. By combining (25) and (26), we can simply write the diagonal blocks of $A$ as a convex combination of automorphisms (of the EJA of symmetric matrices here). We will use the same strategy in the more general setting of EJAs in what follows. This generalization is partly inspired from Example 9 in Gowda (2017), especially for considering the quadratic representation to generalize the operations above of the type $U_1 A U_1$. However, this aforementioned example is not concerned with the diagonal mapping and does not address our question of interest.

**(ii) The diagonal mapping is a convex combination of EJA automorphisms.**[10]

First, we recall the Peirce decomposition theorem which will be crucial in our proof.

---

[10]Note that this step of the proof does not require the fact that the EJA is simple.

**Theorem 15** (Peirce Decomposition in EJAs, e.g. Faraut & Korányi (1994))**.** *Let $\mathcal{J}$ be an EJA with rank $r$ and $\{e_1, \cdots, e_r\}$ a Jordan frame of $\mathcal{J}$. For $i, j \in \{1, \cdots, r\}$, we define the eigenspaces:*

$$\mathcal{J}_{ii} := \{x \in \mathcal{J} : x \circ e_i = x\} = \mathbb{R}\, e_i\,, \tag{27}$$

$$\mathcal{J}_{ij} := \{x \in \mathcal{J} : x \circ e_i = \frac{1}{2}x = x \circ e_j\}, \quad i \neq j\,. \tag{28}$$

*Then the space $\mathcal{J}$ is the orthogonal direct sum of the subspaces $\mathcal{J}_{ij}\, (i \leq j)$. Furthermore,*

$$\mathcal{J}_{ij} \circ \mathcal{J}_{ij} \subseteq \mathcal{J}_{ii} + \mathcal{J}_{jj}\,, \tag{29}$$

$$\mathcal{J}_{ij} \circ \mathcal{J}_{jk} \subseteq \mathcal{J}_{ik} \quad if \quad i \neq k\,, \tag{30}$$

$$\mathcal{J}_{ij} \circ \mathcal{J}_{kl} = \{0\} \quad if \quad \{i, j\} \cap \{k, l\} = \emptyset\,. \tag{31}$$

*Then we can write any $x \in \mathcal{J}$ as:*

$$x = \sum_{1 \leq i \leq j \leq r} x_{ij} = \sum_{i=1}^{r} x_i e_i + \sum_{1 \leq i < j \leq r} x_{ij}\,, \tag{32}$$

*where $x_i \in \mathbb{R}$ and $x_{ij} \in \mathcal{J}_{ij}$. This is the so-called Peirce decomposition of $x \in \mathcal{J}$ associated with the Jordan frame $\{e_1, \cdots, e_r\}$.*

We now define for every $j \in \{2, \cdots, r\}$,

$$\omega_j := e - 2e_j = e_1 + \cdots + e_{j-1} - e_j + e_{j+1} + \cdots + e_r\,, \tag{33}$$

where $e$ is the unit element of the EJA, $e = \sum_{i=1}^{r} e_i$. Observe that $\omega_j^2 = e$ for every $j \in \{1, \cdots, r\}$. We further introduce the quadratic representation $P$ of the EJA (see e.g. section II. 3 in Faraut & Korányi (1994)) which is the map defined for every $\omega \in \mathcal{J}$ by:

$$P_\omega := 2\, L(\omega)^2 - L(\omega^2)\,, \tag{34}$$

where $L(\omega)$ is the linear map of $\mathcal{J}$ defined by $L(\omega)x = \omega \circ x$. Using this definition, it follows that for every $x \in \mathcal{J}$,

$$P_\omega(x) = 2\, \omega \circ (\omega \circ x) - \omega^2 \circ x\,. \tag{35}$$

As for the connection with the case of the algebra of symmetric matrices, notice here that for an associative algebra and a Jordan product $x \circ y = \frac{1}{2}(xy + yx)$, then $P_\omega(x) = \omega x \omega$. We now recursively define a sequence of operators $(\mathcal{C}_k)_{2 \leq k \leq r+1}$ as follows:

$$\mathcal{C}_{r+1}(x) = x, \forall x \in \mathcal{J}\,, \tag{36}$$

$$\mathcal{C}_{k-1}(x) = \frac{1}{2}(\mathcal{C}_k(x) + P_{\omega_{k-1}}(\mathcal{C}_k(x)))\,, \tag{37}$$

$$\forall k \in \{3, \cdots r+1\}, \forall x \in \mathcal{J}\,. \tag{38}$$

The following proposition provides a closed form expression of the pinching operations defined above. In the case of the EJA of symmetric matrices, each pinching operation $\mathcal{C}_k$ sets the last row and column coefficients of the symmetric matrix (except the one on the diagonal) to zero, see (25)-(26) for an illustration.

**Proposition 16.** $\forall r \geq 2, \forall l \in \{-1, 0, \cdots, r-2\}, \forall x \in \mathcal{J}, \mathcal{C}_{r-l}(x) = x - \sum_{k=r-l}^{r} \sum_{1 \leq i < k} x_{ik}\,.$

*Proof.* We prove this result by induction on $l$. The base case follows from observing that for $l = -1$, we have $\mathcal{C}_{r-l}(x) = \mathcal{C}_{r+1}(x) = x$ for all $x \in \mathcal{J}$ by definition of $\mathcal{C}_{r+1}$ in (36). Suppose now that the result holds for some $l \in \{-1, 0, \cdots, r-3\}$:

$$\mathcal{C}_{r-l}(x) = x - \sum_{k=r-l}^{r} \sum_{1 \leq i < k} x_{ik}\,. \tag{39}$$

By the recursive definition of the operator $\mathcal{C}_{r-(l+1)}$, we have for every $x \in \mathcal{J}$,

$$\mathcal{C}_{r-(l+1)}(x) = \frac{1}{2}\left(\mathcal{C}_{r-l}(x) + P_{\omega_{r-(l+1)}}(\mathcal{C}_{r-l}(x))\right), \tag{40}$$

where we recall that $\omega_{r-(l+1)} = e - 2e_{r-(l+1)}$. Since by definition,

$$P_{\omega_{r-(l+1)}}(\mathcal{C}_{r-l}(x)) = 2\omega_{r-(l+1)} \circ (\omega_{r-(l+1)} \circ \mathcal{C}_{r-l}(x)) - \mathcal{C}_{r-l}(x),$$

we have in view of (40) that for every $x \in \mathcal{J}$,

$$\mathcal{C}_{r-(l+1)}(x) = \omega_{r-(l+1)} \circ (\omega_{r-(l+1)} \circ \mathcal{C}_{r-l}(x)). \tag{41}$$

We now compute $\mathcal{C}_{r-(l+1)}(x)$ starting with the following term:

$$\omega_{r-(l+1)} \circ \mathcal{C}_{r-l}(x) \stackrel{(a)}{=} \omega_{r-(l+1)} \circ \left(x - \sum_{k=r-l}^{r}\sum_{1\leq i<k} x_{ik}\right) \tag{42}$$

$$\stackrel{(b)}{=} \omega_{r-(l+1)} \circ \left(\sum_{i=1}^{r} x_i e_i + \sum_{i<j} x_{ij} - \sum_{k=r-l}^{r}\sum_{1\leq i<k} x_{ik}\right) \tag{43}$$

$$\stackrel{(c)}{=} \omega_{r-(l+1)} \circ \left(\sum_{i=1}^{r} x_i e_i + \sum_{k=2}^{r}\sum_{1\leq i<k} x_{ik} - \sum_{k=r-l}^{r}\sum_{1\leq i<k} x_{ik}\right) \tag{44}$$

$$\stackrel{(d)}{=} (e - 2e_{r-(l+1)}) \circ \left(\sum_{i=1}^{r} x_i e_i + \sum_{k=2}^{r-(l+1)}\sum_{1\leq i<k} x_{ik}\right) \tag{45}$$

$$\stackrel{(e)}{=} \sum_{i=1}^{r} x_i e_i + \sum_{k=2}^{r-(l+1)}\sum_{1\leq i<k} x_{ik} - 2x_{r-(l+1)}e_{r-(l+1)} - 2\sum_{k=2}^{r-(l+1)}\sum_{1\leq i<k} e_{r-(l+1)} \circ x_{ik}, \tag{46}$$

where (a) follows from using the closed form of $\mathcal{C}_{r-l}(x)$ in (39), (b) stems from using the Peirce decomposition of $x$ w.r.t. $\{e_1, \cdots, e_r\}$, (c) from rewriting the second term of the Peirce decomposition, (d) uses the definition of $\omega_{r-(l+1)}$ and simplifies the previous difference of sums. Finally, the last equality uses the distributivity of the Jordan product $\circ$ over the addition in the EJA and the fact that $\{e_1, \cdots, e_r\}$ is a Jordan frame.

We now simplify the last sum in the expression above. Observe that $i < k$ (in particular $i \neq k$) and for $k < r - (l+1)$, we have $\{r-(l+1)\} \cap \{i,k\} = \emptyset$. By the orthogonality properties of the Peirce decomposition (see (31)), it follows that $e_{r-(l+1)} \circ x_{ik} = 0$ for $k < r - (l+1)$. The only remaining term in the sum is the term for $k = r - (l+1)$, i.e,

$$\sum_{k=2}^{r-(l+1)}\sum_{1\leq i<k} e_{r-(l+1)} \circ x_{ik} = \sum_{1\leq i<r-(l+1)} e_{r-(l+1)} \circ x_{i,r-(l+1)}.$$

By definition of $\mathcal{J}_{i,r-(l+1)}$ in the Peirce decomposition (see Theorem 15, (28)), we have $e_{r-(l+1)} \circ x_{i,r-(l+1)} = \frac{1}{2}x_{i,r-(l+1)}$. Combining this with (47) and (46), we obtain

$$\omega_{r-(l+1)} \circ \mathcal{C}_{r-l}(x) = \sum_{i=1}^{r} x_i e_i - 2x_{r-(l+1)}e_{r-(l+1)} + \sum_{k=2}^{r-(l+2)}\sum_{1\leq i<k} x_{ik}. \tag{47}$$

To proceed with our computation, observe now that

$$\omega_{r-(l+1)} \circ (\omega_{r-(l+1)} \circ \mathcal{C}_{r-l}(x)) = \omega_{r-(l+1)} \circ \mathcal{C}_{r-l}(x) - 2e_{r-(l+1)} \circ (\omega_{r-(l+1)} \circ \mathcal{C}_{r-l}(x)). \tag{48}$$

We compute the last term in the previous identity using (47) as follows:

$$
\begin{aligned}
e_{r-(l+1)} \circ \left(\omega_{r-(l+1)} \circ \mathcal{C}_{r-l}(x)\right) &= e_{r-(l+1)} \circ \left(\sum_{i=1}^{r} x_i e_i - 2x_{r-(l+1)} e_{r-(l+1)} + \sum_{k=2}^{r-(l+2)} \sum_{1 \le i < k} x_{ik}\right) \\
&= x_{r-(l+1)} e_{r-(l+1)} - 2x_{r-(l+1)} e_{r-(l+1)} + \sum_{k=2}^{r-(l+2)} \sum_{1 \le i < k} e_{r-(l+1)} \circ x_{ik} \\
&= -x_{r-(l+1)} e_{r-(l+1)},
\end{aligned}
\tag{49}
$$

where the last identity from observing that $e_{r-(l+1)} \circ x_{ik} = 0$ for every $2 \le k \le r - (l+2), 1 \le i < k$ again by the orthogonality properties of the Peirce decomposition (see (31)). Overall, combining (41), (47), (48) and (49), we obtain

$$
\mathcal{C}_{r-(l+1)}(x) = \sum_{i=1}^{r} x_i e_i + \sum_{k=2}^{r-(l+2)} \sum_{1 \le i < k} x_{ik} = x - \sum_{k=r-(l+1)}^{r} \sum_{1 \le i < k} x_{ik}.
\tag{50}
$$

This concludes the proof. $\qquad\square$

**Corollary 17.** *The diagonal mapping coincides with $\mathcal{C}_2$.*

*Proof.* Set $l = r - 2$ in Proposition 16 and recall the Peirce decomposition:

$$
x = \sum_{i=1}^{r} x_i e_i + \sum_{i<j} x_{ij} = \sum_{i=1}^{r} x_i e_i + \sum_{k=2}^{r} \sum_{1 \le i < k} x_{ik}.
\tag{51}
$$

$\qquad\square$

**Proposition 18.** *For all $k \in \{0, \cdots, r-1\}$, $\mathcal{C}_{r+1-k}$ is a convex combination of automorphisms.*

*Proof.* We prove this result by induction. The base case $k = 0$ follows from the fact that $\mathcal{C}_{r+1}$ is defined as the identity operator, see (36). Suppose now that $\mathcal{C}_{r+1-k}$ is a convex combination of automorphisms for some $k \in \{0, \cdots, r-2\}$. Hence, there exist a sequence of nonnegative reals $(\lambda)_{1 \le i \le n}$ s.t. $\sum_{i=1}^{n} \lambda_i = 1$ and a sequence of $n$ EJA automorphisms $(\phi_i)_{1 \le i \le n}$ (where $n$ is an unknown integer here that will not be further specified) s.t. $\mathcal{C}_{r+1-k} = \sum_{i=1}^{n} \lambda_i \phi_i$. Using the recursive definition of $\mathcal{C}_{r+1-(k+1)}$, we have for every $x \in \mathcal{J}$,

$$
\begin{aligned}
\mathcal{C}_{r+1-(k+1)}(x) &= \frac{1}{2}\left(\mathcal{C}_{r+1-k}(x) + P_{\omega_{r+1-(k+1)}}\left(\mathcal{C}_{r+1-k}(x)\right)\right) \\
&= \frac{1}{2}\left(\sum_{i=1}^{n} \lambda_i \phi_i(x) + P_{\omega_{r+1-(k+1)}}\left(\sum_{i=1}^{n} \lambda_i \phi_i(x)\right)\right) \\
&= \sum_{i=1}^{n} \frac{\lambda_i}{2} \phi_i(x) + \sum_{i=1}^{n} \frac{\lambda_i}{2} P_{\omega_{r+1-(k+1)}}\left(\phi_i(x)\right),
\end{aligned}
\tag{52}
$$

where we have used the linearity of the operator $P_{\omega_{r+1-(k+1)}}$. To conclude, note that $P_{\omega_{r+1-(k+1)}}$ is an automorphism by Proposition II.4.4, p. 37 in Faraut & Korányi (1994) since $\omega_{r+1-(k+1)}^2 = e$. Moreover, the composition of two automorphisms is yet another automorphism since the set of EJA automorphisms is a group for the composition. Therefore for every $i \in \{1, \cdots, n\}$, the composition $P_{\omega_{r+1-(k+1)}} \phi_i$ is an automorphism. This concludes the proof. $\qquad\square$

**Corollary 19.** *The diagonal mapping is a convex combination of automorphisms.*

*Proof.* Combining Corollary 17 and Proposition 18 (with $k = r - 1$) gives the result. $\qquad\square$

**(iii) Using joint convexity of the relative entropy and properties of automorphisms.** Using the joint convexity of the relative entropy shown in Corollary 3.4 p. 8 in Faybusovich (2016), we can write:

$$D_\Phi(T(x), T(y)) = D_\Phi\left(\sum_{i=1}^n \lambda_i \phi_i(x), \sum_{i=1}^n \lambda_i \phi_i(y)\right) \le \sum_{i=1}^n \lambda_i D_\Phi(\phi_i(x), \phi_i(y)). \tag{53}$$

We now conclude the proof by showing that $D_\Phi(\phi_i(x), \phi_i(y)) = D_\Phi(x, y)$ for any $1 \le i \le n$. We note that the previous quantity is well-defined as the cone of squares of an EJA is invariant under any automorphism of an EJA. We now use the definition of the relative entropy and the fact that every automorphism is also a doubly stochastic map. In particular it is trace preserving. We have

$$\begin{aligned} D_\Phi(\phi_i(x), \phi_i(y)) &= \mathrm{tr}(\phi_i(x) \circ \ln \phi_i(x) - \phi_i(x) \circ \ln \phi_i(y) + \phi_i(y) - \phi_i(x)) \\ &= \mathrm{tr}(\phi_i(x) \circ \ln \phi_i(x)) - \mathrm{tr}(\phi_i(x) \circ \ln \phi_i(y)) + \mathrm{tr}(\phi_i(y)) - \mathrm{tr}(\phi_i(x)) \\ &= \mathrm{tr}(\phi_i(x) \circ \ln \phi_i(x)) - \mathrm{tr}(\phi_i(x) \circ \ln \phi_i(y)) + \mathrm{tr}(y) - \mathrm{tr}(x). \end{aligned} \tag{54}$$

We now show that for any $x \in \mathcal{K}$, and any automorphism $\phi$ of the EJA,

$$\ln \phi(x) = \phi(\ln x). \tag{55}$$

To see this, observe that there exists a Jordan frame $\{q_1, \cdots, q_r\}$ s.t. $x = \sum_{i=1}^r \lambda_i(x) q_i$ where $\{\lambda_i(x)\}_{1 \le i \le r}$ are the eigenvalues of $x$. Then by definition $\ln x = \sum_{i=1}^r \ln(\lambda_i(x)) q_i$. By linearity of $\phi$ we have $\phi(\ln x) = \sum_{i=1}^r \ln(\lambda_i(x)) \phi(q_i)$. Since $\phi$ is an algebra automorphism, it maps Jordan frames to Jordan frames (see e.g. Gowda (2017)). Therefore, $\{\phi(q_1), \cdots, \phi(q_r)\}$ is also a Jordan frame. As a consequence, we can also write $\phi(\ln x) = \sum_{i=1}^r \ln(\lambda_i(y)) \phi(q_i)$. We have shown that $\phi(\ln(x)) = \ln(\phi(x))$.

We now conclude the proof by combining (54) with the identity we have just established by observing that:

$$\mathrm{tr}(\phi_i(x) \circ \ln \phi_i(y)) = \mathrm{tr}(\phi_i(x) \circ \phi_i(\ln(y))) = \mathrm{tr}(\phi_i(x \circ \ln y)) = \mathrm{tr}(x \circ \ln y), \tag{56}$$

where the first equality follows from (55), the second by definition of an automorphism and the last by the fact that an automorphism is trace preserving. Similarly, we have $\mathrm{tr}(\phi_i(x) \circ \ln \phi_i(x)) = \mathrm{tr}(x \circ \ln x)$.

We have shown that $D_\Phi(\phi_i(x), \phi_i(y)) = D_\Phi(x, y)$ and hence from (53) the desired inequality.

Some comments are in order regarding the proof of this theorem.

- **About the particular case of the Jordan spin algebra $\mathbb{L}^n = \mathbb{R} \times \mathbb{R}^{n-1}, n \ge 3$.** We provide some comments about the proof of Theorem 14 when the EJA is the Jordan spin algebra, the limitations of this proof and difficulties to generalize it to (even simple) EJAs. This motivates our specific treatment focused on the properties of the diagonal mapping. The diagonal operator $T$ is a doubly stochastic mapping, i.e, a linear map which is positive, unital (maps the unit element to the unit element) and trace preserving. We refer the reader to e.g. Gowda (2017) for more background and example 7 therein for the aforementioned fact. Then we can use Theorem 9, p. 57 in Gowda (2017) stating that $\mathrm{DS}(\mathbb{L}^n) = \mathrm{conv}(\mathrm{Aut}(\mathbb{L}^n))$. Here conv denotes the convex hull, $\mathrm{Aut}(\mathbb{L}^n)$ is the set of automorphisms of $\mathbb{L}^n$, a (Jordan) algebra automorphism $\phi : \mathcal{J} \to \mathcal{J}$ being an invertible linear transformation satisfying $\phi(x \circ y) = \phi(x) \circ \phi(y)$ for any $x, y \in \mathcal{J}$ and $\mathrm{DS}(\mathbb{L}^n)$ is the set of doubly stochastic mappings. Using this result together with the fact that $T \in \mathrm{DS}(\mathbb{L}^n)$, it follows that $T \in \mathrm{conv}(\mathrm{Aut}(\mathbb{L}^n))$ and hence there exist a family of nonnegative scalars $\{\lambda_i\}_{1 \le i \le n}$ s.t. $\sum_{i=1}^n \lambda_i = 1$ and a family of automorphisms $\{\phi_i\}_{1 \le i \le n}$ of the EJA s.t. $T = \sum_{i=1}^n \lambda_i \phi_i$. The rest of the proof follows the same lines as the proof for the general case of EJAs using joint convexity of the relative entropy.

- The fact that $\mathrm{DS}(\mathbb{L}^n) = \mathrm{conv}(\mathrm{Aut}(\mathbb{L}^n))$ only holds for the Jordan spin algebra. For instance it is not true for the simple algebra of all $3 \times 3$ complex Hermitian matrices (see Example 11 in Gowda (2017); see also example 10 therein for another counterexample). There is a weaker pointwise version for any *simple* EJA, i.e., $\mathrm{DS}(\mathcal{J})z = \mathrm{conv}(\mathrm{Aut}(\mathcal{J}))z$ for any *simple* EJA $\mathcal{J}$ and any $z \in \mathcal{J}$, see Theorem 6 in Gowda (2017). This result is not sufficient for our purposes to invoke the *joint* convexity of the relative entropy as this

would only give different convex decompositions for $T(x), T(y)$ depending on $x, y \in \mathcal{J}$. Nevertheless, we only need a decomposition of the diagonal mapping as a convex combination of automorphisms and this is precisely the result we show in the proof above with a decomposition of the mapping which does not depend on the point we evaluate the diagonal map in.

$\square$

## D  Proofs for Section 5.2

### D.1  Proof of Theorem 10

The proof follows similar lines as the proof of Theorem 4 in Syrgkanis et al. (2015). We provide here a complete proof for our setting of symmetric cone games under Assumption 1. First, it follows from Proposition 5 that the regret $r_i(T)$ of each player $i \in \mathcal{N}$ satisfies

$$r_i(T) \leq \frac{R_i}{\eta} + \eta \sum_{t=1}^{T} \|m_i^t - m_i^{t-1}\|_*^2 - \frac{1}{4\eta} \sum_{t=1}^{T} \|x_i^t - x_i^{t-1}\|^2. \tag{57}$$

The main task now consists in controlling the second term in the above bound using our smoothness assumption and use the negative term. Using assumption 1 and the definition of the payoff vectors, we have

$$\|m_i^t - m_i^{t-1}\|_* = \|m_i(x^t) - m_i(x^{t-1})\|_* \leq L_i \|x^t - x^{t-1}\| \leq L_i \sum_{j=1}^{N} \|x_j^t - x_j^{t-1}\|, \tag{58}$$

where the last inequality follows from Lemma 25 (note that the last norm is the norm defined on the EJA $\mathcal{J}_j$). Then it follows from the bound (58) and Jensen's inequality that

$$\|m_i^t - m_i^{t-1}\|_*^2 \leq N L_i^2 \sum_{j=1}^{N} \|x_j^t - x_j^{t-1}\|^2. \tag{59}$$

We observe in this bound that the payoff vector variations of player $i$ are bounded by the strategy variations of all the players. Therefore to control this error we will be summing up the regret bounds over all players in view of using the negative terms induced by each player in the individual regret bounds to balance the error. Summing up (57) and using (59), we obtain

$$\sum_{i=1}^{N} r_i(T) \leq \frac{\sum_{i=1}^{N} R_i}{\eta} + \left( \eta N \sum_{i=1}^{N} L_i^2 - \frac{1}{4\eta} \right) \sum_{i=1}^{N} \sum_{t=1}^{T} \|x_i^t - x_i^{t-1}\|^2. \tag{60}$$

Setting $\eta = 1/(2\sqrt{N \sum_{i=1}^{N} L_i^2})$ concludes the proof.

### D.2  Min-Max Games

**Theorem 20.** *Consider a two-player symmetric cone game setting ($N = 2$). Assume that the game is zero-sum ($u_2 = -u_1 = f$). Then the following statements hold:*

- *The min-max theorem holds, i.e.* $\displaystyle \min_{x \in \Delta_{\mathcal{K}_1}} \max_{y \in \Delta_{\mathcal{K}_2}} f(x, y) = \max_{y \in \Delta_{\mathcal{K}_2}} \min_{x \in \Delta_{\mathcal{K}_1}} f(x, y) := v^*.$

- *The duality gap is bounded by the sum of regrets, i.e. for any sequences $(x^t), (y^t)$, the average sequences $(\bar{x}_T), (\bar{y}_T)$ defined by $\bar{x}_T = \frac{1}{T} \sum_{t=1}^{T} x^t, \bar{y}_T = \frac{1}{T} \sum_{t=1}^{T} y^t$ satisfy for $T \geq 1$,*

$$0 \leq d(\bar{x}_T, \bar{y}_T) := \max_{y \in \Delta_{\mathcal{K}_2}} f(\bar{x}_T, y) - \min_{x \in \Delta_{\mathcal{K}_1}} f(x, \bar{y}_T) \leq \frac{r_1(T) + r_2(T)}{T}, \tag{61}$$

*where $r_1, r_2$ are the regrets incurred by players 1 and 2 respectively.*

- *The suboptimality gaps for the min and max dual problems are bounded by the sum of regrets, i.e.*
  $d_1(\bar{x}_T) := \max_{y \in \Delta_{\mathcal{K}_2}} f(\bar{x}_T, y) - v^*$ *and* $d_2(\bar{y}_T) := v^* - \min_{x \in \Delta_{\mathcal{K}_1}} f(x, \bar{y}_T)$ *satisfy:*

$$0 \le d_1(\bar{x}_T) \le \frac{r_1(T) + r_2(T)}{T}, \quad 0 \le d_2(\bar{y}_T) \le \frac{r_1(T) + r_2(T)}{T}. \tag{62}$$

- *The point* $(\bar{x}_T, \bar{y}_T) \in \Delta_{\mathcal{K}_1} \times \Delta_{\mathcal{K}_2}$ *is an* $\frac{r_1(T)+r_2(T)}{T}$*-approximate saddle point, i.e. for every* $(x, y) \in \Delta_{\mathcal{K}_1} \times \Delta_{\mathcal{K}_2}$,

$$f(\bar{x}_T, y) - \frac{r_1(T) + r_2(T)}{T} \le f(\bar{x}_T, \bar{y}_T) \le f(x, \bar{y}_T) + \frac{r_1(T) + r_2(T)}{T}. \tag{63}$$

*Proof.* ($i$) follows from the generalized von Neumann min-max theorem upon noticing that $\Delta_{\mathcal{K}_1}, \Delta_{\mathcal{K}_2}$ are convex compact sets and $f$ is convex w.r.t. its first argument and concave w.r.t. its second argument, see e.g. section 5 in Freund & Schapire (1999) for an online learning algorithmic proof. We then have the chain of inequalities:

$$v^* - \frac{r_2(T)}{T} = \min_{x \in \Delta_{\mathcal{K}_1}} \max_{y \in \Delta_{\mathcal{K}_2}} f(x, y) - \frac{r_2(T)}{T} \tag{64}$$

$$\le \max_{y \in \Delta_{\mathcal{K}_2}} f(\bar{x}_T, y) - \frac{r_2(T)}{T} \tag{65}$$

$$\le \max_{y \in \Delta_{\mathcal{K}_2}} \frac{1}{T} \sum_{t=1}^{T} f(x^t, y) - \frac{r_2(T)}{T}$$

$$= \frac{1}{T} \sum_{t=1}^{T} f(x^t, y^t)$$

$$= \min_{x \in \Delta_{\mathcal{K}_1}} \frac{1}{T} \sum_{t=1}^{T} f(x, y^t) + \frac{r_1(T)}{T}$$

$$\le \min_{x \in \Delta_{\mathcal{K}_1}} f(x, \bar{y}_T) + \frac{r_1(T)}{T} \tag{66}$$

$$\le \max_{y \in \Delta_{\mathcal{K}_2}} \min_{x \in \Delta_{\mathcal{K}_1}} f(x, y) + \frac{r_1(T)}{T}$$

$$= v^* + \frac{r_1(T)}{T}. \tag{67}$$

Comparing the expressions in lines (65) and (66) immediately gives the bound on the duality gap in ($ii$), while comparing (65) with (67) and (64) with (66) give the bounds on the suboptimality gap in ($iii$).

Finally, comparing (65) and (66) again (or just taking the result in ($ii$)) tells us that

$$\max_{y \in \Delta_{\mathcal{K}_2}} f(\bar{x}_T, y) - \frac{r_2(T)}{T} \le \min_{x \in \Delta_{\mathcal{K}_1}} f(x, \bar{y}_T) + \frac{r_1(T)}{T}. \tag{68}$$

Since the right-hand-side of (68) is a lower bound to $f(\bar{x}_T, \bar{y}_T) + \frac{r_1(T)}{T}$, we have

$$\max_{y \in \Delta_{\mathcal{K}_2}} f(\bar{x}_T, y) - \frac{r_1(T) + r_2(T)}{T} \le f(\bar{x}_T, \bar{y}_T), \tag{69}$$

i.e. that $\bar{y}_T$ is an $\frac{r_1(T)+r_2(T)}{T}$-approximate best response to $\bar{x}_T$. On the other hand, since the left-hand-side of (68) is an upper bound to $f(\bar{x}_T, \bar{y}_T) - \frac{r_2(T)}{T}$ we have

$$f(\bar{x}_T, \bar{y}_T) \le \min_{x \in \Delta_{\mathcal{K}_1}} f(x, \bar{y}_T) + \frac{r_1(T) + r_2(T)}{T}, \tag{70}$$

i.e. that $\bar{x}_T$ is an $\frac{r_1(T)+r_2(T)}{T}$-approximate best response to $\bar{y}_T$. Putting (69) and (70) together gives us $(iv)$ (i.e. that $(\bar{x}_T, \bar{y}_T)$ is an $\frac{r_1(T)+r_2(T)}{T}$-approximate saddle point). $\qquad \square$

## E    More General Hybrid Symmetric Cone Min-Max Games

We consider here bilinear min-max games over the most generalized simplexes (OSCMWU) can handle. Since bilinear utilities can be characterized by linear functions over the tensor product space, we express the min-max problem using tensor product notation for symmetry between players:

$$\min_{x \in \Delta_{\mathcal{K}_1}} \max_{y \in \Delta_{\mathcal{K}_2}} \langle z, x \otimes y \rangle \ , \tag{71}$$

where $\mathcal{K}_1 = \mathbb{S}_+^{n_1} \times \prod_{i=1}^{m_1} \mathbb{L}_+^{d_i}$ is the cone squares of the EJA $\mathcal{J}_1 = \mathbb{S}^{n_1} \times \prod_{i=1}^{m_1} \mathbb{L}^{d_{1,i}}$ and $\mathcal{K}_2 = \mathbb{S}_+^{n_2} \times \prod_{i=1}^{m_2} \mathbb{L}_+^{d_{2,i}}$ is the cone squares of the EJA $\mathcal{J}_1 = \mathbb{S}^{n_2} \times \prod_{i=1}^{m_2} \mathbb{L}^{d_{2,i}}$. $z \in \mathcal{J}_1 \otimes \mathcal{J}_2$ is the "game tensor", and similarly to the case of quantum games we can obtain the gradients of the players' utilities with respect to their own strategies: $m_1(x,y) = -\operatorname{tr}_2 \big( z \circ (e_1 \otimes y) \big), m_2(x,y) = \operatorname{tr}_1 \big( z \circ (x \otimes e_2) \big)$ where $e_1, e_2$ are the identity elements of $\mathcal{J}_1, \mathcal{J}_2$ respectively. See appendix below for a derivation of this and the definition of the partial trace functions $\operatorname{tr}_1, \operatorname{tr}_2$. Then, if the utilities are bounded, i.e., $\max_{x \in \Delta_{\mathcal{K}_1}, y \in \Delta_{\mathcal{K}_2}} |\langle z, x \otimes y \rangle| \leq L$ for some $L > 0$, then we have the Lipschitz constants $L_1, L_2 \leq L$. Assumption 1 holds (by Proposition 23 using Assumption 21 which is satisfied, see Appendix F below for more details). Finally, we have $R_1 = \ln(\operatorname{rank}(\mathcal{J}_1)) = \ln(n_1 + 2m_1)$ and similarly that $R_2 = \ln(n_2 + 2m_2)$, so by Theorem 11, in order to obtain an $\epsilon$-saddle point of (71) it suffices for both players to run (OSCMWU) with stepsize $\eta = \frac{1}{2L}$ for $T \geq \frac{4(\ln(n_1+2m_1)+\ln(n_2+2m_2))L}{\epsilon}$ iterations.

## F    Games with Multilinear Utilities

In this section we present some technical results specific to games with multilinear utilities.

### F.1    Smoothness of payoff vectors for multilinear games

In our multilinear game setting, we follow the notation of the main part but replace Assumption 1 with the following assumption on the utilities $u_i : \mathcal{X} \to \mathbb{R}$:

**Assumption 21.** *Each payoff function $u_i : \mathcal{X} = \prod_j \mathcal{J}_j \to \mathbb{R}$ is multilinear and satisfies $|u_i(x)| \leq L_i \ \forall x \in \mathcal{X}$.*

The bound on each $|u_i(x)|$ in Assumption 21 has a natural interpretation that the payoffs that each player can obtain are bounded. Proposition 23 allows us to convert this into a lipschitz bound on the payoff vectors. We first have the following technical lemma:

**Lemma 22.** *Let $\mathcal{K}_1, \ldots, \mathcal{K}_n$ be the cones of squares of EJAs $\mathcal{J}_1, \ldots, \mathcal{J}_n$ respectively, and suppose that $u_i : \prod_{j=1}^n \mathcal{J}_j \to \mathbb{R}$ is multilinear with $|u_i(x)| \leq L_i \ \forall x \in \prod_{j=1}^n \Delta_{\mathcal{K}_j}$. Then for any $j \in [n]$ the linear function $u_i(\cdot, x_{-j}) : \mathcal{J}_j \to \mathbb{R}$ satisfies*

$$\|u_i(\cdot, x_{-j})\|_{\operatorname{tr},1,*} \leq L_i.$$

*Proof.* This follows directly from Lemma 26:

$$\|u_i(\cdot, x_{-j})\|_{\operatorname{tr},1,*} = \max_{\|x_j'\|_{\operatorname{tr},1}=1} |u_i(x_j', x_{-j})| = \max_{x_j' \in \Delta_{\mathcal{K}_j}} |u_i(x_j', x_{-j})| \leq L_i.$$

$\qquad \square$

This then gives us the following bound on $\|m_i^t - m_i^{t-1}\|_*$ in terms of $\sum_{j \neq i} \|x_j^t - x_j^{t-1}\|$.

**Proposition 23.** *Suppose that the game satisfies Assumption 21 and is played repeatedly so that at time $t$ player $i$'s strategy is given by $x_i^t$. Fix the norm $\|\cdot\| = \|\cdot\|_{\operatorname{tr},1}$. Then for each player $i$ and time $t$, we have the following bound:*

$$\|m_i^t - m_i^{t-1}\|_* \leq L_i \sum_{j \neq i} \|x_j^t - x_j^{t-1}\|.$$

*Proof.* We first observe that, for given $i \neq j$ and $\forall x_i \in \Delta_{\mathcal{K}_i}$, $x_{-ij} \in \prod_{k \notin \{i,j\}} \Delta_{\mathcal{K}_k}$, and time $t$,

$$\left| u_i(x_i, x_j^t, x_{-ij}) - u_i(x_i, x_j^{t-1}, x_{-ij}) \right| \leq L_i \left\| x_j^t - x_j^{t-1} \right\| \tag{72}$$

by Lemma 22 and Assumption 21. We thus have

$$
\begin{aligned}
\left\| u_i(\cdot, x_j^t, x_{-ij}) - u_i(\cdot, x_j^{t-1}, x_{-ij}) \right\|_* &= \max_{\|x_i\|=1} \left| u_i(x_i, x_j^t, x_{-ij}) - u_i(x_i, x_j^{t-1}, x_{-ij}) \right| \\
&= \max_{x_i \in \Delta_{\mathcal{K}_i}} \left| u_i(x_i, x_j^t, x_{-ij}) - u_i(x_i, x_j^{t-1}, x_{-ij}) \right| \\
&\leq L_i \left\| x_j^t - x_j^{t-1} \right\|,
\end{aligned}
$$

where the second equality follows from Lemma 26 and the fact that $u_i$ is multilinear, and the inequality follows from (72). Finally, index the $j \neq i$ by $j_1, j_2, \ldots, j_{n-1}$ and use $x_{j_{[l,k]}}^t$ for $l \leq k$ to denote the tuple $(x_{j_l}^t, \ldots, x_{j_k}^t)$. We have

$$\left\| m_i^t - m_i^{t-1} \right\|_* = \left\| u_i(\cdot, x_{-i}^t) - u_i(\cdot, x_{-i}^{t-1}) \right\|_* \leq \sum_{l=1}^{n-1} L_i \left\| x_{j_l}^t - x_{j_l}^{t-1} \right\| = L_i \sum_{j \neq i} \left\| x_j^t - x_j^{t-1} \right\|,$$

where the inequality follows from Lemma 22. $\qquad \square$

## F.2 Choi map for EJAs

Utility functions $u : \mathcal{J}_1 \times \mathcal{J}_2 \to \mathbb{R}$ that are multilinear on the product/direct-sum space can also be characterized by linear functions on the tensor product space, i.e.,

$$
\begin{aligned}
\tilde{u} &: \mathcal{J}_1 \otimes \mathcal{J}_2 \to \mathbb{R}, \\
x \otimes y &\mapsto \langle z, x \otimes y \rangle
\end{aligned}
\tag{73}
$$

for some 'game tensor' $z \in \mathcal{J}_1 \otimes \mathcal{J}_2$. (We write this section for two players for simplicity of notation, but it can easily be extended to the multiplayer case.)

To obtain from this characterization the gradient of $u$ with respect to, say, the $x$-player, we use an extension of the Choi–Jamiołkowski isomorphism, which corresponds quantum states with quantum channels, to EJAs. The Choi–Jamiołkowski isomorphism has been similarly used to compute gradients in quantum games, see e.g, Vasconcelos et al. (2023).

To do this, define the linear map (which depends on the game tensor $z$ defined in (73)) by

$$
\begin{aligned}
\Theta_z &: \mathcal{J}_2 \to \mathcal{J}_1, \\
y &\mapsto \mathrm{tr}_2 \left( z \circ (e_1 \otimes y) \right)
\end{aligned}
\tag{74}
$$

where the partial trace function on $\mathcal{J}_1 \otimes \mathcal{J}_2$ with respect to $\mathcal{J}_2$ is defined by

$$
\begin{aligned}
\mathrm{tr}_2 &: \mathcal{J}_1 \otimes \mathcal{J}_2 \to \mathcal{J}_1, \\
x \otimes y &\mapsto \mathrm{tr}(y)x
\end{aligned}
\tag{75}
$$

and $e_1$ is the identity element in $\mathcal{J}_1$. The following proposition then says that the gradient of a utility function $u$ (characterized by game tensor $z$ according to (73)) with respect to $x$ is simply $\Theta_z(y)$.

**Proposition 24.** *Given a game tensor $z \in \mathcal{J}_1 \otimes \mathcal{J}_2$, define $\Theta_z$ as in (74). Then the following holds:*

$$\langle z, x \otimes y \rangle = \langle x, \Theta_z(y) \rangle. \tag{76}$$

*Proof.* Let $v_1, \ldots, v_{r_1}$ be a basis for $\mathcal{J}_1$ and $w_1, \ldots, w_{r_2}$ be a basis for $\mathcal{J}_2$. Then $\{v_i \otimes w_j\}_{i \in [r_1], j \in [r_2]}$ is a basis for $\mathcal{J}_1 \otimes \mathcal{J}_2$, and we can write

$$z = \sum_{ij} z_{ij}(v_i \otimes w_j)$$

for some $z_{ij} \in \mathbb{R}$. We then have

$$
\begin{aligned}
\langle x, \Theta_z(y) \rangle &= \langle x, \mathrm{tr}_2 \left( z \circ (e_1 \otimes y) \right) \rangle \\
&= \left\langle x, \mathrm{tr}_2 \left( \sum_{ij} z_{ij}(v_i \otimes w_j) \circ (e_1 \otimes y) \right) \right\rangle \\
&= \left\langle x, \mathrm{tr}_2 \left( \sum_{ij} z_{ij} \big( v_i \otimes (w_j \circ y) \big) \right) \right\rangle \\
&= \left\langle x, \sum_{ij} z_{ij} \mathrm{tr}_2 \left( v_i \otimes (w_j \circ y) \right) \right\rangle \\
&= \left\langle x, \sum_{ij} z_{ij} \mathrm{tr}(w_j \circ y) v_i \right\rangle \\
&= \sum_{ij} z_{ij} \langle v_i, x \rangle \langle w_j, y \rangle \\
&= \sum_{ij} z_{ij} \langle v_i \otimes w_j, x \otimes y \rangle \\
&= \left\langle \sum_{ij} z_{ij} v_i \otimes w_j, x \otimes y \right\rangle \\
&= \langle z, x \otimes y \rangle .
\end{aligned}
$$

$\square$

## G    Technical Lemmas

**Lemma 25.** *Let $\mathcal{J} := \prod_{i=1}^N \mathcal{J}_i$ be the product Euclidean Jordan Algebra of $n$ EJAs $\mathcal{J}_i$ and let $\| \cdot \|_{\mathcal{J}_i}$ be the norm induced by the EJA inner product $\langle \cdot, \cdot \rangle_{\mathcal{J}_i}$ on each one of the EJA $\mathcal{J}_i$, i.e. for $x = (x_1, \cdots, x_N) \in \mathcal{J}, \|x\|_{\mathcal{J}}^2 = \sum_{i \in \mathcal{N}} \|x_i\|_{\mathcal{J}_i}^2$ where $\|x_i\|_{\mathcal{J}_i}^2 = \langle x_i, x_i \rangle_{\mathcal{J}_i}$. Then for every $x = (x_1, \cdots, x_N) \in \mathcal{J}, \|x\|_{\mathcal{J}} \leq \sum_{i=1}^N \|x_i\|_{\mathcal{J}_i}$.*

*Proof.* An immediate proof stems from observing that $\|x\|_{\mathcal{J}}^2 = \sum_{i \in \mathcal{N}} \|x_i\|_{\mathcal{J}_i}^2 \leq \left( \sum_{i \in \mathcal{N}} \|x_i\|_{\mathcal{J}_i} \right)^2$ where the last inequality follows from noticing that all the terms in the left hand side are contained in the expansion of the square in the right hand side of the inequality.

Here is an alternative proof using the fact that $\| \cdot \|_{\mathcal{J}}$ is a norm. Using the triangular inequality and the definition of the inner product on the product of EJAs:

$$
\begin{aligned}
\|x\|_{\mathcal{J}} = \|(x_1, \cdots, x_N)\|_{\mathcal{J}} &= \|(x_1, 0, \cdots, 0) + (0, x_2, 0, \cdots, 0) + \cdots + (0, \cdots, 0, x_N)\|_{\mathcal{J}} \\
&\leq \|(x_1, 0, \cdots, 0)\|_{\mathcal{J}} + \|(0, x_2, 0, \cdots, 0)\|_{\mathcal{J}} + \cdots + \|(0, \cdots, 0, x_N)\|_{\mathcal{J}} \\
&= \sum_{i=1}^N \|x_i\|_{\mathcal{J}_i} .
\end{aligned}
\tag{77}
$$

$\square$

**Lemma 26.** *Let $\mathcal{J}$ be an EJA and $\mathcal{K}$ its cone of squares. If $h : \mathcal{J} \to \mathbb{R}$ is a linear function, then we have*

$$
\max_{\|x\|_{\mathrm{tr},1}=1} |h(x)| = \max_{x \in \Delta_{\mathcal{K}}} |h(x)|.^{[11]}
$$

[11]Maxima are attained since $|h(x)|$ is continuous and both $\{x \in \mathcal{J} : \|x\|_{\mathrm{tr},1} = 1\}$ and $\Delta_{\mathcal{K}}$ are compact.

*Proof.* Let $v^* := \max_{\|x\|_{\mathrm{tr},1}=1} |h(x)|$. Since $\Delta_{\mathcal{K}} \subseteq \{x \in \mathcal{J} : \|x\|_{\mathrm{tr},1} = 1\}$, we need only prove that $\max_{x \in \Delta_{\mathcal{K}}} |h(x)| \geq v^*$.

Note that $|h(x)| = \max\{h(x), -h(x)\}$, so

$$v^* = \max_{\|x\|_{\mathrm{tr},1}=1} \max\{h(x), -h(x)\} = \max\left\{\max_{\|x\|_{\mathrm{tr},1}=1} h(x), \max_{\|x\|_{\mathrm{tr},1}=1} -h(x)\right\}. \tag{78}$$

Without loss of generality we can assume that $v^* = \max_{\|x\|_{\mathrm{tr},1}=1} h(x)$, since if it is instead the case that $v^* = \max_{\|x\|_{\mathrm{tr},1}=1} -h(x)$ then we could simply have taken $-h$ as our linear function instead.[12] The maximum of $h(x)$ over $\{x \in \mathcal{J} : \|x\|_{\mathrm{tr},1} = 1\}$ is attained at some $x^*$; let $\sum_i \lambda_i q_i$ be its spectral decomposition. Since $\lambda_i = |\lambda_i| \operatorname{sign}(\lambda_i)$, we have

$$x^* = \sum_i \lambda_i q_i = \sum_i |\lambda_i| \operatorname{sign}(\lambda_i) q_i,$$

i.e., $x^*$ is a convex combination of $\{\operatorname{sign}(\lambda_i)q_i\}_i$ (since $\sum_i |\lambda_i| = 1$). Then, since each $\operatorname{sign}(\lambda_i)q_i$ also lies in $\{x \in \mathcal{J} : \|x\|_{\mathrm{tr},1} = 1\}$ and $x^*$ is the maximizer of the linear function $h(x)$ over $\{x \in \mathcal{J} : \|x\|_{\mathrm{tr},1} = 1\}$, we have

$$v^* = h(x^*) = h(\operatorname{sign}(\lambda_i)q_i), \qquad \forall i : |\lambda_i| \neq 0.$$

Now pick any $i : |\lambda_i| \neq 0$. $q_i \in \Delta_{\mathcal{K}}$ and satisfies $|h(q_i)| = |h(\operatorname{sign}(\lambda_i)q_i)| = v^*$. Therefore we have $\max_{x \in \Delta_{\mathcal{K}}} |h(x)| \geq |h(q_i)| = v^*$. $\qquad\square$

---

[12]i.e., define $g(x) := -h(x)$. We then want to show that $\max_{\|x\|_{\mathrm{tr},1}=1} |g(x)| = \max_{x \in \Delta_{\mathcal{K}}} |g(x)|$, and we are in the case that $v^* := \max_{\|x\|_{\mathrm{tr},1}=1} |g(x)| = \max_{\|x\|_{\mathrm{tr},1}=1} g(x)$.

