# OpenReview forum: "Optimistic Online Learning in Symmetric Cone Games"
_TMLR — Accepted by TMLR_

### Review · Reviewer_YWN9 · 2025-11-30

**Summary Of Contributions:**

This paper develops a unified framework called Symmetric Cone Games (SCGs) for online learning over any generalized simplex (the trace-one slice of a symmetric cone), which has a wide range of applications in various settings like normal-form games and quantum games. To compute the approximate Nash equilibrium of any two-player zero-sum SCG, the authors proposed an online algorithm called Optimistic Symmetric Cone Multiplicative Weights Updates (OSCMWU) based on the Optimistic Follow-the-Regularized-Leader (OFTRL) framework, where the Symmetric Cone Negative Entropy (SCNE) is added as a regularizer. By using the strong convexity of the SCNE with respect to the trace-one norm, which is newly proved in this paper, the authors have shown that OSCMWU can achieve an $\widetilde{O}(\epsilon^{-1})$ complexity for computing $\epsilon$-saddle points.

**Additional Comments:**

NA

**Audience:**

Yes

**Audience Explanation:**

Overall, this paper studies and proposes an interesting framework for online learning over generalized simplexes. Regarding the theoretical results proved in the paper, to the best of the reviewer's knowledge, this is the first work that proves the strong convexity of the Symmetric Cone Negative Entropy (SCNE) with respect to the trace-one norm. Hence, this work will be of interest to TMLR's audience.

**Broader Impact Concerns:**

Since this paper is mainly a theoretical work in online learning, the reviewer doesn't think there is any related concern on the work's ethical implications.

**Claims And Evidence:**

Yes

**Claims Explanation:**

The reviewer finds that detailed proofs are provided for all theoretical results presented in this paper. However, the paper's quality can be potentially improved by adding a few numerical experiments to justify the effectiveness of the proposed OSCMWU algorithm.

**Requested Changes:**

Below is a list of changes that the authors need to take into consideration:

(1) The reviewer wonders whether it is appropriate to describe the proposed algorithm (OSCMWU) as being "projection-free". A more description might be that the algorithm doesn't involve any Euclidean projection onto the symmetric cone. However, under the OFTRL framework, the proposed algorithm does contain an implicit Bregman projection via the exponential-and-normalize step. Therefore, the authors might need to consider whether the description needs to be slightly modified here.

(2) Regarding the applicability of Assumption 1, would it be possible for the authors to briefly discuss how the Lipschitz constants scale like in the Examples (normal-form games, PSD matrix games, quantum games) of the SCGs?

---

> ### Author Response · Authors · 2026-01-17
> **Rebuttal**
>
> We thank the reviewer for their time and valuable feedback. We reply to their comments and questions below.
>
> > The reviewer finds that detailed proofs are provided for all theoretical results presented in this paper. However, the paper's quality can be potentially improved by adding a few numerical experiments to justify the effectiveness of the proposed OSCMWU algorithm.
>
> While our main contributions are theoretical, we have now added 3 illustrative numerical experiments in the two applications we consider (distance metric learning and facility location/Fermat–Weber), comparing SCMWU and OSCMWU and showing the applicability of OSCMWU to different SCGs. These simulations include: (a) a distance metric learning instance (application 1, see section 6.1 p. 10), (b) a facility location instance (application 2, see section 6.2 p. 11) and (c ) an online variant of facility location to highlight the online aspect (see section 6.2 p. 12). New added text appears in blue color in the revised manuscript.
>
>
> > (1) The reviewer wonders whether it is appropriate to describe the proposed algorithm (OSCMWU) as being "projection-free". A more description might be that the algorithm doesn't involve any Euclidean projection onto the symmetric cone. However, under the OFTRL framework, the proposed algorithm does contain an implicit Bregman projection via the exponential-and-normalize step. Therefore, the authors might need to consider whether the description needs to be slightly modified here.
>
> Indeed, by `projection-free', we mean that the algorithm does not involve any Euclidean projection onto the symmetric cone. We agree that the proposed algorithm does contain an 'implicit' Bregman projection via the exponential-and-normalize step. We have removed all occurences of the adjective 'projection-free' in the manuscript for clarity and added the following more precise textual description once: 'does not require any Euclidean projection onto the symmetric cone'. Our main point is that this normalized exponential map computation step exploits the generalized simplex structure. We provide more details regarding this explicit computation depending on the symmetric cone under consideration in remark 3 p. 7 and appendix B.4 p. 18-19.
>
> > (2) Regarding the applicability of Assumption 1, would it be possible for the authors to briefly discuss how the Lipschitz constants scale like in the Examples (normal-form games, PSD matrix games, quantum games) of the SCGs?
>
> The Lipschitz constant typically scales with a uniform bound on the utilities of each player in most of the settings we describe. This is the case for instance for normal-form games as shown in Syrgkanis et al. 2015, Theorem 4 (who suppose that utilities are bounded by 1). For PSD matrix games, the smoothness constant scales with the linear operator norm when utilities are linear operators. For instance, we derive the Lipschitz constants for a simplex-spectraplex game in section 6.1 p. 9-10 and show that $L_1 = L_2 = \max_i \|A_i\|_{\text{tr},\infty}$ where $A_i$ are the matrices defining the min-max problem. For quantum games, we show in appendix F, Proposition 22 that Assumption 1 is satisfied with $L_i$ being a uniform bound on the payoffs for multilinear games.

---

### Review · Reviewer_3Twb · 2025-12-06

**Summary Of Contributions:**

This paper formulates a class of multi-player games named symmetric cone games (SCGs), where each player's strategy lies in a generalized simplex. Furthermore, they provide several examples included in SCGs. Particularly for two-player zero-sum SCGs, this paper proposes an online learning algorithm called optimistic symmetric cone multiplicative weights updates (OSCMWU). For the regret analysis of the proposed method, the authors provide the result on the strong convexity of the symmetric cone negative entropy with respect to the trace-one norm.

**Additional Comments:**

- The authors claim that the proposed method is projection-free and its updates are closed-form. Although I think that these are benefits regarding computational efficiency, there is no experimental validation of this benefit. Furthermore, the authors note that existing methods have computational complexity almost identical to that of the proposed method for two specific applications, as discussed in Section 6. Therefore, I currently understand that the merit of the proposed framework is only the versatility and ability to support online learning.
- To be honest, I felt that the construction of Sections 4 and 5 was strange. First, since their titles are similar, I could not imagine any difference in the contents of these sections from the titles alone. Second, although Section 4 starts with the sentence ``we present our ... algorithm,'' the regret analysis is also partially discussed in Section 4 (mainly in Section 5). I think the construction, where Sections 4 and 5 focus on the proposed algorithm and regret analysis, respectively, is easier to follow.
- Why didn't you provide the experimental result? Is there any limitation regarding computational complexity? (I ask this since ``developing scalable implementations'' is listed as future work.)

**Audience:**

Yes

**Audience Explanation:**

The problems discussed in this paper relate to a wide range of applications, as discussed in Section 3.
Therefore, I believe that there is an audience who is interested in this line of study.

**Claims And Evidence:**

Yes

**Claims Explanation:**

Although I could not read most of the proof in the appendix due to time limitations, the equations and analyses appear rigorous.

However, I felt the discussion of related work was slightly unclear, particularly regarding [1], though this relates to the main contribution of this paper. I believe that discussing the difference from [1] about the following is required:
- Regarding the strong convexity shown in Theorem 7, Lemma 2.1 of [1], which is partially from [2], shows a related result for the (strict) convexity. I think that the difference should be explicitly discussed.
- The problem setup itself is similar. However, this similarity with [1] is discussed only briefly in Section 4 (and the appendix). I believe that a more comprehensive comparison is required regarding, e.g., the differences in the scope of the problem setup and the above convexity analysis, though the improvement in the regret analysis has already been discussed.

[1] Ilayda Canyakmaz, Wayne Lin, Georgios Piliouras, Antonios Varvitsiotis, Multiplicative Updates for Online Convex Optimization over Symmetric Cones

[2] Jein-Shan Chen · Shaohua Pan, An entropy-like proximal algorithm and the exponential multiplier method for convex symmetric cone programming

Post rebuttal --------

My concern is resolved by the revision.

**Requested Changes:**

- Please add the discussion for the related work described above.
- In Section 1, the following sentence is written. I think that ``applying OSCMWU'' can cause confusion since it can be read as the authors will show experimental results. Please consider rephrasing this expression.
>We illustrate the versatility of our framework by applying OSCMWU to two representative SCGs, showcasing its generality, simplicity, and robust convergence across different cone geometries.

- In Eq. (3), there is $u\_i^t$. Is this a typographical error of $u_i$? Please fix or add a definition of this.

- Regarding Example 1 in Section 3.1, there is an abuse of notation with respect to $u_i$. Moreover, I think that the function of $a\_i$ and $x\_{- i}$, $u\_i (a\_i, x\_{-i})$, is not defined. Please add the explanation.

- Example 2 is not informative. Please add mathematical definitions or citations, or please delete it.

- Many examples are shown in Sections 3.1 and 3.2. Do these examples satisfy the assumption of the proposed method's analysis? Or, do these examples just demonstrate the broadness of the problem definition? Anyway, please consider explicitly describing this point.

- Regarding the paragraph right after Theorem 7, I could not understand the message of this paragraph since a similar statement is discussed in the following paragraph. Please consider rephrasing these two successive paragraphs.

### Minor points
- There are several inconsistent uses of font type. In particular, subscripts in $\| \cdot \|\_{\rm tr}$ are often italic. Please recheck the entire paper.
- In the first paragraph of Section 3.2, ``machine learning'' is capitalized. Please consider making it lowercase.

---

> ### Author Response · Authors · 2026-01-17
> **Rebuttal**
>
> We thank the reviewer for their time and constructive feedback.
>
> **Comparison to [1] and [2].** We provide more elements below:
> - Strong convexity:  [2, Lemma 3.2.c] and [1, Lemma 2.1 (ii)] only state that the SCNE is strictly convex on the symmetric cone. We show that the SCNE is strongly convex with respect to the trace 1 norm, which is a stronger result. We have added this discussion and the reference [2] in the new remark 9 in p. 8.
> - Setting: The online learning setting we consider is similar as mentioned in section 4. However, [1] does not consider the general setting of SCGs. More importantly their analysis does not exploit the predictability of the game setting in contrast to the fully adversarial setting. We exploit the game structure to obtain our improved rate using optimism. Our analysis builds on the OFTRL framework and strong convexity of SCNE rather than standard regret analysis of MWU [1]. We have added more details in p. 10. The applications we consider in sections 3.1 and 3.2 are also different.
>
> **Requested Changes:**
> > Please add the discussion for the related work described above.
>
> Please see the revision, remark 9 p. 8 and beginning of p. 10.
>
> > ``applying OSCMWU'' can cause confusion [...] Please consider rephrasing this expression.
>
> We have added a few simulations (see section 6) and we hence keep the formulation.
>
> > In Eq. (3), there is $u_i^t$. Is this a typographical error of $u_i$?
>
> Typo corrected, it should be $u_i$.
>
> > Example 1 in Section 3.1, there is an abuse of notation with respect to $u_i$. [...] function of $a_i$ and $x_{-i}$, $u_i(a_i,x_{-i})$, is not defined.
>
> We have added a footnote for clarification in example 1 (p. 5) and the notation $u_i(a_i,x_{-i}) := \sum_{a_{-i} \in \mathcal{A}_{-i}} x_{a_{-i}} u_i(a_i, a_{-i})$ and $\mathcal{A}_{-i} := \prod_{j \in \mathcal{N}, j \neq i} \mathcal{A}_j\,.$
>
> > Example 2 is not informative. Please add mathematical definitions or citations, or please delete it.
>
> We have expanded the discussion of example 2 (p. 5) with notation, concrete examples and references. The goal of example 2 is to illustrate the use of second-order cones as an important class of symmetric cones within our SCG framework.
>
> > [Examples in Sections 3.1 and 3.2] Do these examples satisfy the assumption of the proposed method's analysis? [...]
>
> Yes, our analysis applies to these examples. We now clarify this in the end of p. 4.
>
> Our analysis mainly relies on the smoothness assumption 1 discussed in p. 4. Section 3.1 mentions it is satisfied for example 1, section 6 that it is for the examples of section 3.2 and appendix F shows that it also holds in multilinear games with bounded utilities.
>
> > paragraph right after Theorem 7, [...] Please consider rephrasing these two successive paragraphs.
>
> Revised for clarity. The goal of these paragraphs is to provide more insights regarding the proof technique and its novelty for the strong convexity result.
>
> **Minor points:** Fixed now.
>
> **Additional comments:**
>
> > the proposed method is projection-free and its updates are closed-form. [...]
>
> - We use 'projection-free' and 'closed-form' as algorithmic descriptors. We have removed the adjective 'projection-free' for clarity.
> - We propose a general framework applicable across all SCGs, including online learning settings in which offline methods described in prior work cannot be immediately applied. As noted by the reviewer, the main merits of the algorithmic framework are (i) applicability to any instance of SCGs and (ii) online learning support with provable guarantees. To stress this point, we have added an online variant of the facility location application in section 6.2 with a few simulations. Additionally, our guarantee depends only logarithmically on the rank of the underlying EJA rather than scaling with the ambient dimension.
>
> > the construction, where Sections 4 and 5 focus on the proposed algorithm and regret analysis, respectively, is easier to follow.
>
> We now separate the presentation of the algorithm (section 4, up to Proposition 4) from the analysis (now in section 5), renaming section 5 and adding subsections to section 5. The content remains unchanged.
>
> > Why didn't you provide the experimental result? Is there any limitation regarding computational complexity? [...]
>
> - **Simulations.** While the main contributions and focus of this work are theoretical, we have now added 3 illustrative simulations (see section 6).
> - **Computational complexity.** As we discuss in remark 3 p. 7 and appendix B.4, it depends on the exponential computation which itself depends on the symmetric cone. This computation is straightforward for the nonnegative orthant and the SOC whereas it can be more expensive for the PSD cone (as it might require an eigenvalue decomposition). In the PSD case, our algorithm could also be enhanced by using randomization techniques to scale to large dimensions. We leave this future work. We have revised the end of the conclusion.

---

> > ### Comment · Reviewer_3Twb · 2026-01-18
> >
> > I appreciate the detailed feedback.
> > The authors have addressed all of my concerns.
> > In particular, though I did not recommend it, I felt that the paper's quality improved with the addition of numerical experiments.

---

> > > ### Author Response · Authors · 2026-01-18
> > >
> > > We appreciate the prompt feedback and we thank the reviewer again for their time and useful input which helped us improve the manuscript.

---

### Review · Reviewer_Bggv · 2026-01-13

**Summary Of Contributions:**

This paper introduces a unified framework for symmetric cone games, in which players’ strategies lie in the generalized simplex of a symmetric cone. This formulation subsumes several well-studied game classes, including normal-form games, positive semidefinite (PSD) matrix games, and quantum games.

To solve these games, the authors propose a single online, projection-free algorithm — OSCMWU —which can be viewed as a variant of the Optimistic Follow-the-Regularized-Leader method. The convergence analysis hinges on establishing strong convexity of the negative entropy map over symmetric cones. This step is technically demanding and relies heavily on tools from Euclidean Jordan algebra. Under smooth convex utilities, the authors derive a convergence rate on the order of $1/\varepsilon$, which explicitly reflects the complexity of the strategy space through the rank of the Euclidean Jordan algebra associated with the cone. In particular, the a logarithmic dependency on the rank is achieved.

**Strengths and Weaknesses**

Strengths :
1. The paper is fairly well-written and easy to follow.
2. The unifying framework is of practical interest and the existence of a general algorithm which specializes along various classes of symmetric cone games is an important contribution.

Weaknesses:
1. The only two applications considered are already tackled by Nesterov smoothing with similar performances. It would be interesting to have alternative applications that stress the “online” aspect of the algorithm proposed.

**Audience:**

Yes

**Audience Explanation:**

I believe this paper is of general interest as it offers a unifying framework for solving a diverse class of games using the same general method. The rates derived are able to capture the complexity of the ambiant strategy space (via the rank of the EJA associated to the symmetric cone).

**Broader Impact Concerns:**

No concerns.

**Claims And Evidence:**

Yes

**Claims Explanation:**

The main technical contribution of the paper is the establishment of the strong convexity of the negative entropy over general symmetric cones. The proof proceeds by lower bounding the negative entropy between two elements by the entropy of their respective projections onto a Jordan frame (the projection is called a Diagonal operator). The authors then show that the negative entropy between these projected elements can itself be lower bounded by the trace-1 norm of the difference between the original elements. The key projection inequality relies on a nice inductive argument, which decomposes the Diagonal operator into a convex combination of automorphisms that commute with the logarithmic map associated with the underlying EJA. These developments are technically sound and accurate. The other contributions (regret bounds and complexity rate) follow from standard argument from the online convex optimization literature.

**Requested Changes:**

Here are minor changes to address :
* Page 2 : Optimistic Symmetric Cone Multiplicative Weights “Update”
* Page 3 : Does the proof of Baes (2006) present the same constants regarding the strong convexity of the negative entropy. Is so, this should be stated.
* Page 22 - Theorem 14 : The Jordan frame $\{e_1,…, e_r\}$ is introduced twice, making the statements slightly confusing.
* Equations (41) and (49) : $i < j$
* Page 24 - Below equation (50) : $w_{r+1 - (k+1)} ^2 = e$

---

> ### Author Response · Authors · 2026-01-17
> **Rebuttal**
>
> We thank the reviewer for their careful reading, accurate summary of our contributions, and constructive feedback. We reply to the reviewer's comments below.
>
> > Weaknesses: The only two applications considered are already tackled by Nesterov smoothing with similar performances. It would be interesting to have alternative applications that stress the “online” aspect of the algorithm proposed.
>
> Thank you for the suggestion. We have now added to the revised manuscript an online variant of the facility location application to highlight this aspect. We consider a sequential setting in which demands arrive over time (streaming demand) and the decision maker must update the facility location on the fly. Please see section 6.2 p. 11-12 for a detailed description of the variant and a simple illustrative simulation comparing SCMWU to OSCMWU on a synthetic instance of the problem.
>
> **Requested changes**
>
> We have implemented all the required changes in the manuscript. More details below.
>
> > Here are minor changes to address:
>
> > Page 2 : Optimistic Symmetric Cone Multiplicative Weights “Update”
>
> This is fixed now in the revised version.
>
> > Page 3 : Does the proof of Baes (2006) present the same constants regarding the strong convexity of the negative entropy. Is so, this should be stated.
>
> Yes, the strong convexity constant is the same (equal to 1) if the reviewer is referring to p.8. As suggested, we have now revised the manuscript adding 'leading to the same strong convexity constant (one) as in Theorem 7' after 'Baes (2006)' in p. 8.
>
> > Page 22 - Theorem 14 : The Jordan frame $e_1, \cdots, e_r$ is introduced twice, making the statements slightly confusing.
>
> Indeed. We have now removed the second occurence of the Jordan frame from the statement to avoid any confusion.
>
> > Equations (41) and (49) : $i < j$
>
> Thank you for spotting these typos which are now fixed in the revised version.
>
> > Page 24 - Below equation (50) : $\omega^2_{r+1-(k+1)} = e$
>
> Indeed there was a missing square, thank you for spotting this, the typo is corrected now.

---

### Author Response · Authors · 2026-01-17
**General Author Comments and Revised Manuscript**

We thank all the reviewers for their time and valuable feedback which helped us improve our manuscript. The detailed individual responses to each one of the reviewers can be found after each review. We briefly summarize below the main changes in the new uploaded revised manuscript (main new additions in blue color) in response to the reviewers' comments and suggestions:

- As suggested by reviewer 3Twb, sections 4 and 5 have been slightly restructured to separate the presentation of the algorithm (section 4) from the analysis (which is now presented separately in section 5). Section 5 has been renamed 'Regret Analysis and Average Iterate Convergence Guarantees' instead of the previous title 'Optimistic Online Learning in Symmetric Cone Games' and subsection titles (5.1 to 5.3) have been added for more structure. The presentation order and the content remain unchanged.

- We have added remark 9 to contrast strong convexity of SCNE w.r.t. the trace one norm with the strict convexity result (with the provided reference) as requested by reviewer 3Twb. We have also slightly expanded the technical comparison to Canyakmaz et al. in the beginning of p. 10.

- We have expanded the description of example 2 in section 3.1 to describe convex games with $\ell_2$ ball strategy sets as an illustration of SCGs with second-order cones.

- In section 6.2, we have added an online variant of the facility location application to highlight the online aspect as suggested by reviewer Bggv (see section 6.2 p. 12).

- **Simulations.** While the main contributions and focus of this work are theoretical, we have now added 3 simulations for illustration to compare SCMWU and OSCMWU on the 2 applications we discuss in the paper (see figures 1 to 3): (a) a distance metric learning instance (application 1, see section 6.1 p. 10), (b) a facility location instance (application 2, see section 6.2 p. 11) and (c ) an online variant of facility location to highlight the online aspect as suggested by reviewer Bggv (see section 6.2 p. 12). The code to reproduce the experiments on Google Colab (jupyter notebooks) is available in the supplementary material.

- **Typos and other minor requested changes.** All the spotted typos have now been fixed and we have implemented all the requested changes by the reviewers in the revised manuscript.

---

### Decision · Action_Editor_h5YV · 2026-02-25

**Recommendation:** Accept as is

**Audience:**

Yes

**Audience Explanation:**

This paper studies an interesting framework around online learning over generalizations of the simplex; this is hence interesting to individuals working in online learning or learning in games.

**Claims And Evidence:**

Yes

**Claims Explanation:**

The paper introduce a class of matrix games: symmetric cone games (SCGs) which forms a unifying framework for normal-form games (simplex strategies), quantum games (density matrices), and continuous games with ball-constrained strategies. The Optimistic Multiplicative Weights Update's method is shown to be implementable in this setting, analyzed thoroughly, and illustrated numerically. In particular, the authors show that the symmetric cone negative entropy is strongly convex with respect to the trace-one norm as a key helper result. The reviewers were overall positive about the writing and developments, especially with the added numerical illustrations.